# Cancer burden in Nepal, 1990–2017: An analysis of the Global Burden of Disease study

Gambhir Shrestha[ID][1]*, Rahul Kumar Thakur[ID][2], Rajshree Singh[ID][3], Rashmi Mulmi[4], Abha Shrestha[5], Pranil Man Singh Pradhan[1]

1 Department of Community Medicine, Maharajgunj Medical Campus, Institute of Medicine, Tribhuvan University, Maharajgunj, Kathmandu, Nepal, 2 Department of Internal Medicine, Jacobi Medical Center/ Albert Einstein College of Medicine, Bronx, New York, United States of America, 3 Department of Diagnostic Radiology, Mercy Catholic Medical Center, Darby, Philadelphia, United States of America, 4 Department of Cancer Prevention, Control and Research, B.P. Koirala Memorial Cancer Hospital, Bharatpur, Chitwan, Nepal, 5 Department of Community Medicine, Kathmandu University School of Medical Sciences, Dhulikhel, Kavre, Nepal

* gamvir.stha@gmail.com

**Data Availability Statement:** The datasets used and/or analyzed during the current study are available in the article itself and can be found freely at http://ghdx.healthdata.org/.

## Abstract

### Introduction

Cancer is the second leading cause of death and a major public health problem in the world. This study reports the trend and burden of cancer from 1990 to 2017 along with its risk factors in Nepal.

### Methods

This study used the database of the Institute of Health Metrics and Evaluation's Global Burden of Diseases on cancer from Nepal to describe the most recent data available (2017) and trends by age, gender, and year from 1990 to 2017. The data are described as incidence, prevalence, disability-adjusted life years (DALY), and percentage change.

### Results

In 2017, the age-standardized cancer incidence and mortality rates were 101.8/100,000 and 86.6/100,000 respectively in Nepal. Cancer contributed to 10% of total deaths and 5.6% of total DALYs in Nepal. The most common cancers were the breast, lung, cervical, stomach and oral cavity cancers. The number of new cancer cases and deaths in Nepal have increased from 1990 to 2017 by 92% and 95% respectively. On the other hand, age-standardized incidence and mortality rates decreased by 5% and 7% respectively. The leading risk factors of cancer were tobacco use, dietary factors, unsafe sex, air pollution, drug use, and physical inactivity.

### Conclusions

This study highlighted the burden of cancer in Nepal, contributing to a significant number of new cancer cases, deaths and DALY. A comprehensive approach including prevention,

**Funding:** The author(s) received no specific funding for this work.

**Competing interests:** The authors have declared that no competing interests exist.

early diagnosis and treatment, and rehabilitation should be urgently taken to reduce the burden of cancer.

## Introduction

Cancer is the second leading cause of death worldwide, with 18.1 million new cases and 9.6 million deaths in 2018 [1–3]. The Global Cancer Observatory 2018 estimated the age-standardized cancer incidence and mortality rates to be 103.7/100,000 and 77.8/100,000, respectively, in Nepal [4]. Mortality is high in low- and middle-income countries (LMICs) and associated with poor prognosis which is due to lack of awareness, delayed diagnosis, inequity on health accessibility, and affordability as compared to high-income countries [5, 6].

In addition to the internal risk factors (genetics, ethnicity, and race), and external risk factors such as diet, tobacco and alcohol intake, viral infections, and exposure to chemicals and radiation, the incidence and prevalence of cancer also vary due to geography, socio-economic status, religious or cultural practices [7, 8]. In Nepal, major risk factors include tobacco (smoking and smokeless), betel quid, areca nut, indoor and outdoor air pollution, alcohol, viral infections like Hepatitis B, Hepatitis C, HIV and Human Papilloma Virus, *Helicobacter pylori*, and other dietary habits [9–13]. The dietary habits are largely associated with epidemiological transition, which has changed the lifestyle of people in terms of their habits, social practices, diet, nutrition, surrounding environment.

Former articles unfolding the burden of cancer in Nepal were very restricted in scope because the analysis was limited to short periods, and geographical coverage, incomplete, or limited to special settings like particular hospitals [6, 12–14]. The Global Burden of Diseases (GBD) 2017 study provides a unique opportunity to analyze trends in cancer incidence and mortality that will help in policy planning and resource allocation and effective implementation of preventive and curative measures to curb the burden of cancer in Nepal.

This study aimed to understand the trend of a shift in incidence, prevalence, and mortality using standard parameters and correlate the findings with the status of prevalent risk factors in Nepal. Here, we report a systematic and comprehensive picture of the magnitude and time trends of cancer and estimates of disability over time, by age and gender in Nepal from 1990 to 2017. The findings from this analysis can be a foundation for setting priorities for future research and formulating effective policies based on current evidence.

## Materials and methods

### Study design

This cross-sectional study was based on the systematic analysis of the Institute for Health Metrics and Evaluation (IHME) database to measure the burden of cancer in Nepal [15]. We used data for incidence, prevalence, mortality, and DALY rates for cancer from 1990 to 2017 available from IHME's Global Burden of Disease (GBD) database. The GBD 2017 used surveillance and survey data, published and unpublished papers, vital registration, hospital data to quantify the magnitude of health loss for 354 causes from 195 countries, including Nepal [16].

### Operational definition

In this study, the following measures of disease burden were defined similarly to the IHME database definition.

- Disability-adjusted life years (DALY) is a summary measure that combines time lost through premature death and time lived in states of less than optimal health, loosely referred to as "disability" [17].

- Age-standardized rate is a weighted average of the age-specific rates per 100,000 persons, where the weights are the proportions of persons in the corresponding age groups of the WHO standard population [18].

- Uncertainty interval (UI) is a range of values that is likely to include the correct estimate of disease burden for a given cause. Narrow UI indicates that evidence is strong, while wide UI shows that evidence is weaker.

## Statistical analysis

The data from the IHME database were downloaded, compiled, and analyzed in Microsoft Excel 2013. We then created tables and graphs to relate patterns and trends in mortality, incidence and prevalence rates, and DALYs for different types of cancer in Nepal by age and gender. The explanation of metrics, data gathering procedures, and analytical methods used for GBD 2017 are described elsewhere [16]. The percent change was evaluated to indicate the course and extent of the trends of different types of cancer from 1990 to 2017. A 95% UI was presented to show the strength of the estimates.

## Results

### Incidence

An estimated 22,869 new cancer cases were recorded, out of which 45% were males and 55% were females in 2017. The total number of new cases for all forms of neoplasm increased by 91.77% from 1990 to 2017 but the age-standardized incidence rates (ASIR per 100,000 population) for all forms of cancer in Nepal decreased by 4.79% from 106.9 (92.07–124.49) in 1990 to 101.8 (88.71–116.18) in 2017. Among males, the ASIR (per 100,000 population) of cancer increased from 95.42 (77.86–112.92) in 1990 to 99.69 (80.13–116.84) in 2017. The ASIR (per 100,000 population) for all forms of cancer among females decreased from 118.75 (95.03–151.09) in 1990 to 103.42 (83.34–128.76) in 2017. Table 1 shows the all-age incidence and age-standardized incidence rate of all forms of cancer from 1990 to 2017. Overall, in both sexes, there has been a significant increase in the incidence of pancreatic, ovarian, liver, kidney, and thyroid cancer with a percentage change of 76.76, 66.46, 50.01, 49.87 and 49.32 respectively from 1990 to 2017 in Nepal (Table 1).

The most common sites for cancer among men were lung, stomach, prostate, oral cavity, and colorectal in 2017. Of the five most common cancer sites, prostate and colorectal cancer have an increasing trend of incidence with a percentage change of 47.62 and 32.58 respectively, from 1990 to 2017. There has been a decreasing trend of incidence in males with a percentage change of 7.23, 30.35, and 14.78 for lung, stomach, and oral cavity cancer respectively, from 1990 to 2017 (Table 1).

Among women, breast cancer had the highest ASIR, followed by cervical, lung, colorectal, and stomach cancer in 2017. Of the five common cancers, breast, lung, and colorectal cancer have an increasing trend of incidence with percentage change of 30.01, 2.33, and 9.21, respectively from 1990 to 2017. Cervical and stomach cancer have a decreasing trend of incidence with a percentage change of 51.49 and 32.43 from 1990 to 2017 (Table 1). Table 2 shows an increase in the incidence of cancer with an increase in age. The Age-specific incidence rate is highest among people aged 80 years and older for both males and females being 893.53

**Table 1. Total all-age incidence and age-standardized incidence rates for different types of cancer and their percentage change by gender in Nepal, 1990–2017.**

| Morphology | Total all-Age incidence (95% UI) | | | Age-Standardized Incidence Rate (95% UI), per 100,000 | | |
|---|---|---|---|---|---|---|
| | 1990 | 2017 | Change, % | 1990 | 2017 | Change, % |
| **All Neoplasms** | | | | | | |
| Male | 5155.48(4189.69–6137.9) | 10233.65(8214.79–12131.72) | 98.5 | 95.42(77.86–112.92) | 99.69(80.13–116.84) | 4.47 |
| Female | 6769.93(5388.12–8694.68) | 12635.69(10111.73–15809.03) | 86.64 | 118.75(95.03–151.09) | 103.42(83.34–128.76) | -12.91 |
| Both | 11925.41(10091.48–14179.48) | 22869.35(19691.18–26250.64) | 91.77 | 106.96(92.07–124.49) | 101.84(88.71–116.18) | -4.79 |
| **Tracheal, bronchus, and lung cancer** | | | | | | |
| Male | 705.22(464.57–1168.97) | 1314.63(965.71–2113.84) | 86.41 | 14.02(9.15–23.44) | 13(9.59–20.89) | -7.23 |
| Female | 356.66(107.82–607.35) | 900.1(344.4–1478.07) | 152.37 | 7.82(2.21–13.69) | 8(3.03–13.11) | 2.33 |
| Both | 1061.88(647.61–1622) | 2214.73(1438.65–3084.48) | 108.57 | 11.01(6.57–16.63) | 10.39(6.78–14.37) | -5.68 |
| **Breast cancer** | | | | | | |
| Male | 16.71(12.88–21.34) | 44.55(33.14–56.53) | 166.63 | 0.33(0.25–0.41) | 0.44(0.33–0.56) | 36.15 |
| Female | 931.26(589.22–1530.81) | 2673.65(1895.6–4942.89) | 187.1 | 16.45(10.7–27.53) | 21.41(15.36–39.65) | 30.09 |
| Both | 947.97(603.92–1548.29) | 2718.2(1943.39–4996.42) | 186.74 | 8.23(5.4–13.65) | 11.54(8.33–21.22) | 40.23 |
| **Cervical cancer** | | | | | | |
| Female | 1915.84(1124.14–2606.4) | 1981.22(1342.01–2749.12) | 3.41 | 30.6(17.78–41.11) | 14.84(10.3–20.35) | -51.49 |
| Both | 1915.84(1124.14–2606.4) | 1981.22(1342.01–2749.12) | 3.41 | 15.07(8.77–20.26) | 7.93(5.49–10.91) | -47.35 |
| **Colon and rectum cancer** | | | | | | |
| Male | 267.13(163.38–419.37) | 710.73(489.07–1066.62) | 166.06 | 5.48(3.36–8.54) | 7.27(5.06–10.85) | 32.58 |
| Female | 279.46(168.05–488.46) | 726.79(554.2–939.45) | 160.07 | 6.1(3.74–10.47) | 6.66(5.14–8.56) | 9.21 |
| Both | 546.59(372.5–842.51) | 1437.52(1157.26–1840.79) | 163 | 5.8(4.04–8.81) | 6.96(5.64–8.88) | 20.07 |
| **Esophageal cancer** | | | | | | |
| Male | 341.78(270.6–426.58) | 703.73(546.01–848.99) | 105.9 | 6.52(5.22–8.1) | 6.84(5.33–8.24) | 4.83 |
| Female | 278.23(217.39–364.97) | 358.72(278.03–480.67) | 28.93 | 5.75(4.46–7.62) | 3.17(2.46–4.21) | -44.84 |
| Both | 620.01(523.85–744.75) | 1062.45(872.33–1266.93) | 71.36 | 6.16(5.24–7.45) | 4.91(4.06–5.81) | -20.22 |
| **Lip and oral cavity cancer** | | | | | | |
| Male | 550.83(362.76–832.84) | 897.44(634.4–1186.05) | 62.92 | 10(6.61–14.98) | 8.52(6.05–11.18) | -14.78 |
| Female | 279.15(207.02–366.58) | 663.9(514.04–829.45) | 137.83 | 5.56(4.15–7.32) | 5.64(4.4–7.05) | 1.44 |
| Both | 829.98(610.24–1132.26) | 1561.34(1239.01–1892.42) | 88.12 | 7.84(5.8–10.55) | 7.02(5.6–8.44) | -10.49 |
| **Stomach cancer** | | | | | | |
| Male | 646.41(497.44–823.97) | 895.67(674.81–1128.95) | 38.56 | 12.95(9.9–16.4) | 9.02(6.88–11.27) | -30.35 |
| Female | 478.42(361.92–682.81) | 728.23(569.41–933.44) | 52.22 | 9.43(7.23–13.44) | 6.37(5.01–8.17) | -32.43 |
| Both | 1124.83(932–1366.3) | 1623.9(1360.63–1932.22) | 44.37 | 11.24(9.32–13.54) | 7.63(6.42–9.05) | -32.09 |
| **Bladder cancer** | | | | | | |
| Male | 105.83(76.59–177.49) | 290.54(208.65–501.91) | 174.54 | 2.45(1.77–4.01) | 3.13(2.26–5.36) | 27.77 |
| Female | 52.21(34.57–77.7) | 121.05(85.3–158.66) | 131.84 | 1.21(0.8–1.78) | 1.14(0.8–1.49) | -5.58 |
| Both | 158.04(119.3–230.51) | 411.59(317.64–618.63) | 160.43 | 1.83(1.39–2.61) | 2.07(1.61–3.09) | 12.88 |
| **Brain and nervous system cancer** | | | | | | |
| Male | 152.42(74.07–291.73) | 264.18(127.46–505.58) | 73.32 | 1.94(0.87–4.03) | 2.2(1.07–4.15) | 13.44 |
| Female | 230.67(72.05–430.74) | 244.87(162.78–376.41) | 6.16 | 2.67(0.97–5.08) | 1.82(1.22–2.79) | -31.85 |
| Both | 383.09(227.96–595.94) | 509.05(334.85–831.76) | 32.88 | 2.3(1.59–3.57) | 2(1.34–3.25) | -12.89 |
| **Gallbladder and biliary tract cancer** | | | | | | |
| Male | 81.92(57.88–124.7) | 205.41(125.57–289.95) | 150.75 | 1.76(1.26–2.64) | 2.11(1.3–2.93) | 20.07 |
| Female | 154.6(106.91–313.73) | 394.07(276.1–617.17) | 154.9 | 3.46(2.39–7.14) | 3.57(2.49–5.58) | 3.14 |
| Both | 236.52(176.07–407.94) | 599.48(450.87–808.87) | 153.47 | 2.6(1.95–4.48) | 2.88(2.16–3.91) | 11.03 |
| **Hodgkin lymphoma** | | | | | | |
| Male | 131.39(86.11–215.6) | 89.51(51.17–179.06) | -31.88 | 1.74(1.15–2.92) | 0.73(0.41–1.47) | -58.04 |
| Female | 57.39(38.41–82.72) | 56.25(32.36–105) | -1.99 | 0.74(0.51–1.04) | 0.38(0.22–0.7) | -48.29 |

*(Continued)*

**Table 1.** (Continued)

| Morphology | Total all-Age incidence (95% UI) | | | Age-Standardized Incidence Rate (95% UI), per 100,000 | | |
|---|---|---|---|---|---|---|
| | 1990 | 2017 | Change, % | 1990 | 2017 | Change, % |
| Both | 188.78(131.64–285.14) | 145.76(88.31–268.3) | -22.79 | 1.24(0.89–1.86) | 0.55(0.34–1) | -55.93 |
| **Kidney cancer** | | | | | | |
| Male | 77.54(45.2–136.02) | 241.37(146.34–407.07) | 211.31 | 1.19(0.69–1.87) | 2.18(1.33–3.67) | 83.64 |
| Female | 83.1(47.03–153.1) | 175.08(113.16–266.04) | 110.7 | 1.16(0.67–1.97) | 1.37(0.89–2.08) | 18.25 |
| Both | 160.63(99.43–254.13) | 416.46(277.6–630.08) | 159.26 | 1.17(0.72–1.68) | 1.76(1.18–2.65) | 49.87 |
| **Larynx cancer** | | | | | | |
| Male | 335.18(255.16–435.7) | 521.26(392.53–700.53) | 55.52 | 6.3(4.81–8.11) | 4.91(3.74–6.52) | -22.07 |
| Female | 155.21(106.4–207.22) | 213.28(144.75–278.59) | 37.41 | 2.96(1.94–3.93) | 1.78(1.21–2.33) | -39.72 |
| Both | 490.39(390.74–602.55) | 734.54(589.52–886.81) | 49.79 | 4.68(3.74–5.74) | 3.28(2.66–3.96) | -29.96 |
| **Leukemia** | | | | | | |
| Male | 325.91(231.63–441.24) | 460.4(308.03–624.23) | 41.27 | 4.16(3.02–5.51) | 3.93(2.66–5.22) | -5.55 |
| Female | 337.66(203.54–652.13) | 452.88(328.85–603.89) | 34.12 | 4.26(2.85–7.08) | 3.38(2.47–4.46) | -20.62 |
| Both | 663.56(453.75–1047.37) | 913.28(707.33–1143.76) | 37.63 | 4.21(3.26–5.59) | 3.64(2.88–4.5) | -13.41 |
| **Liver cancer** | | | | | | |
| Male | 137.33(78.94–199.14) | 479.48(269.1–835.9) | 249.14 | 2.59(1.47–3.68) | 4.62(2.61–7.99) | 78.14 |
| Female | 81.4(42.96–117.07) | 218.29(152.78–307.08) | 168.16 | 1.67(0.9–2.38) | 1.94(1.37–2.71) | 16.42 |
| Both | 218.74(156.91–292.31) | 697.77(481.13–1097.59) | 219 | 2.15(1.55–2.86) | 3.22(2.24–5.04) | 50.01 |
| **Malignant skin melanoma** | | | | | | |
| Male | 12.87(8.21–23.59) | 27.01(17.56–44.73) | 109.9 | 0.23(0.15–0.4) | 0.26(0.17–0.42) | 12.63 |
| Female | 10.05(5.09–25.2) | 28.97(17.26–61.81) | 188.11 | 0.19(0.1–0.47) | 0.24(0.14–0.51) | 22.88 |
| Both | 22.92(15.09–43.61) | 55.98(37.62–96.55) | 144.2 | 0.21(0.14–0.39) | 0.25(0.17–0.43) | 17.23 |
| **Mesothelioma** | | | | | | |
| Male | 10(5–20.2) | 26.24(16.84–38.94) | 162.43 | 0.19(0.1–0.38) | 0.25(0.17–0.37) | 34.47 |
| Female | 12.16(4.77–24.28) | 16.84(10.4–25.83) | 38.45 | 0.22(0.09–0.42) | 0.14(0.09–0.21) | -37.08 |
| Both | 22.16(11.35–42.79) | 43.08(29.79–61.96) | 94.38 | 0.2(0.11–0.38) | 0.19(0.13–0.27) | -5.19 |
| **Multiple myeloma** | | | | | | |
| Male | 34.09(20.19–58.15) | 95.3(58.31–164.69) | 179.53 | 0.66(0.4–1.14) | 0.92(0.57–1.57) | 38.57 |
| Female | 36.99(24.52–59.16) | 115.8(76.52–184.39) | 213.1 | 0.8(0.54–1.31) | 1.04(0.69–1.63) | 28.95 |
| Both | 71.08(50.15–110.77) | 211.1(149.69–319.86) | 196.99 | 0.73(0.53–1.14) | 0.98(0.7–1.48) | 33.96 |
| **Nasopharynx cancer** | | | | | | |
| Male | 88.94(60.45–133.99) | 144.83(103.02–198.51) | 62.84 | 1.47(1.05–2.07) | 1.3(0.94–1.74) | -11.49 |
| Female | 79.39(48.39–134.49) | 89.31(57.52–137.26) | 12.49 | 1.22(0.81–1.84) | 0.67(0.46–0.99) | -44.88 |
| Both | 168.33(122.96–232.6) | 234.14(175.31–304.21) | 39.09 | 1.35(1.03–1.77) | 0.97(0.75–1.24) | -28.05 |
| **Non-Hodgkin lymphoma** | | | | | | |
| Male | 125.02(79.57–191.96) | 268.68(183.84–399.43) | 114.91 | 2.01(1.26–2.89) | 2.48(1.7–3.64) | 23.25 |
| Female | 77.69(53.76–108.98) | 202.85(121.62–293.74) | 161.11 | 1.3(0.9–1.75) | 1.67(1.01–2.4) | 28.16 |
| Both | 202.7(156.93–264.57) | 471.53(356.32–601.59) | 132.62 | 1.66(1.3–2.06) | 2.05(1.56–2.6) | 23.74 |
| **Non-melanoma skin cancer** | | | | | | |
| Male | 130.95(79.52–261.89) | 267.67(180.27–486.29) | 104.4 | 2.43(1.79–3.98) | 2.6(1.93–4.11) | 6.99 |
| Female | 82.03(27.64–218.67) | 167.65(68.28–423.69) | 104.36 | 1.2(0.59–2.74) | 1.19(0.6–2.72) | -0.27 |
| Both | 212.98(106.81–480.6) | 435.31(249.91–913.86) | 104.39 | 1.81(1.19–3.35) | 1.84(1.21–3.35) | 1.57 |
| **Other pharynx cancer** | | | | | | |
| Male | 229.16(132.04–363.16) | 575.13(389.77–813.69) | 150.97 | 4.13(2.36–6.58) | 5.35(3.65–7.52) | 29.42 |
| Female | 142.06(105.49–194.52) | 319.83(241.07–449.91) | 125.14 | 2.67(1.97–3.68) | 2.65(2–3.74) | -0.85 |
| Both | 371.22(274.7–519.59) | 894.96(685.86–1168.91) | 141.09 | 3.43(2.52–4.81) | 3.95(3.04–5.12) | 15 |
| **Ovarian cancer** | | | | | | |

*(Continued)*

**Table 1.** (Continued)

| Morphology | Total all-Age incidence (95% UI) | | | Age-Standardized Incidence Rate (95% UI), per 100,000 | | | |
|---|---|---|---|---|---|---|---|
| | 1990 | 2017 | Change, % | 1990 | 2017 | Change, % |
| Female | 138.26(92.52–247.88) | 481.95(350.35–635.29) | 248.58 | 2.47(1.7–4.28) | 3.82(2.81–5.03) | 54.5 |
| Both | 138.26(92.52–247.88) | 481.95(350.35–635.29) | 248.58 | 1.21(0.83–2.1) | 2.02(1.49–2.65) | 66.46 |
| **Pancreatic cancer** | | | | | | |
| Male | 77.34(49.37–123.56) | 286.61(184.84–458.52) | 270.58 | 1.59(1.02–2.52) | 2.91(1.89–4.59) | 82.76 |
| Female | 56.67(37.15–77.43) | 244.45(162.11–327.68) | 331.37 | 1.3(0.82–1.79) | 2.26(1.5–3.01) | 73.09 |
| Both | 134.01(91.93–187) | 531.06(368.56–742.66) | 296.28 | 1.45(0.98–2.03) | 2.57(1.79–3.56) | 76.76 |
| **Prostate cancer** | | | | | | |
| Male | 216.09(154.06–292.1) | 773.84(546.95–1020.69) | 258.11 | 6.09(4.22–8.27) | 8.98(6.38–11.83) | 47.62 |
| Both | 216.09(154.06–292.1) | 773.84(546.95–1020.69) | 258.11 | 3.03(2.11–4.12) | 4.15(2.93–5.46) | 36.97 |
| **Testicular cancer** | | | | | | |
| Male | 47.47(29.26–67.03) | 41.89(23.94–61.29) | -11.76 | 0.6(0.38–0.84) | 0.32(0.18–0.46) | -47.21 |
| Both | 47.47(29.26–67.03) | 41.89(23.94–61.29) | -11.76 | 0.29(0.18–0.41) | 0.14(0.08–0.21) | -50.98 |
| **Thyroid cancer** | | | | | | |
| Male | 32.83(22.94–45.88) | 94.32(67.22–129.51) | 187.26 | 0.54(0.38–0.74) | 0.84(0.6–1.14) | 54.25 |
| Female | 119.84(75.57–204.24) | 347.42(231.98–558.61) | 189.9 | 1.74(1.13–3) | 2.45(1.67–3.92) | 40.7 |
| Both | 152.67(105.81–235.3) | 441.74(319.98–638.42) | 189.33 | 1.14(0.82–1.74) | 1.71(1.26–2.44) | 49.32 |
| **Uterine cancer** | | | | | | |
| Female | 124.08(71.23–173.81) | 266.19(187.93–360.81) | 114.53 | 2.49(1.48–3.47) | 2.24(1.6–3.03) | -9.92 |
| Both | 124.08(71.23–173.81) | 266.19(187.93–360.81) | 114.53 | 1.21(0.72–1.69) | 1.17(0.84–1.59) | -3.25 |

(738.62–1032.70) and 605.72 (483.22–738.17) respectively. Fig 1 shows the change in crude incidence rate from 53.10 (43.16–63.22) to 71.59 (57.46–84.86) in males and from 70.05 (55.75–89.96) to 81.02 (64.84–101.37) in females from 1990 to 2017.

## Prevalence

In 2017, the total number of estimated prevalent cancer cases was 58,570 with an overall prevalence of 29% among males and 71% among females. The age-standardized prevalence rates (ASPR) per 100,000 population of all forms of cancer increased by 3.43% from 243.75 (203.7–299.21) in 1990 to 252.12 (209.89–318.24) in 2017. The ASPR for both sexes was highest for breast cancer followed by cervical, colorectal and prostate cancer. Among males, the ASPR (per 100,000 population) for all cancers was 139.21 in 1990 and 163.13 in 2017 with the five most common cancer sites being prostate, oral, colorectal, larynx, and stomach cancer. Similarly, among females, the ASPR (per 100,000 population) for all cancers was 350.94 in 1990 and 328.98 in 2017 with the highest prevalence of breast cancer followed by cervical, oral cavity, colorectal, and thyroid cancer (Table 3).

## Mortality

Cancer accounted for 10% of total deaths in Nepal in 2017, with the major contribution being from lung cancer (1.3%). There were an estimated 18,315 deaths from cancer, and of all deaths, 51% were in males and 49% in females in Nepal in 2017. The total number of deaths for all forms of neoplasm increased by 94.72% from 1990 to 2017. The age-standardized mortality rate (ASMR) per 100,000 population for cancer decreased by 7.14% from 93.21 (81–107.54) in 1990 to 86.56 (75.59–97.03) in 2017. The ASMR per 100,000 population for all forms of cancer

**Table 2. Age-specific number of new cancer cases, deaths, incidence rate and mortality rates of cancer by gender in Nepal, 2017.**

| Age group | Total deaths (95%UI) | Age-specific death rates, Number in 100,000 (95%UI) | Total new cases, (95%UI) | Age-specific incidence rates, Number in 100,000 (95%UI) |
|---|---|---|---|---|
| **All Ages** | | | | |
| Male | 9294.53(7463.08–11079.92) | 65.02(52.21–77.51) | 10233.65(8214.79–12131.72) | 71.59(57.46–84.86) |
| Female | 9020(7265.11–11017.63) | 57.84(46.58–70.64) | 12635.69(10111.73–15809.03) | 81.02(64.84–101.37) |
| Both | 18314.54(16013.37–20596.07) | 61.27(53.57–68.9) | 22869.35(19691.18–26250.64) | 76.51(65.88–87.82) |
| **1 to 4** | | | | |
| Male | 41.75(19.22–65.53) | 3.32(1.53–5.22) | 82.3(39.13–126.28) | 6.55(3.12–10.06) |
| Female | 32.07(15.43–51.23) | 2.7(1.3–4.31) | 63.55(30.86–99.76) | 5.34(2.59–8.39) |
| Both | 73.82(35.8–110.95) | 3.02(1.46–4.54) | 145.85(71.64–214.52) | 5.96(2.93–8.77) |
| **5 to 9** | | | | |
| Male | 55.94(34.61–86.6) | 3.54(2.19–5.47) | 86.31(54.39–130.01) | 5.46(3.44–8.22) |
| Female | 41.47(24.37–62.44) | 2.75(1.62–4.14) | 65.21(38.83–96.74) | 4.32(2.57–6.41) |
| Both | 97.41(66.18–139.76) | 3.15(2.14–4.52) | 151.52(105.14–210.21) | 4.9(3.4–6.8) |
| **10 to 14** | | | | |
| Male | 57.33(37.01–83.62) | 3.44(2.22–5.01) | 75.04(48.82–105.22) | 4.5(2.93–6.31) |
| Female | 44.68(28.56–64.07) | 2.76(1.76–3.96) | 64.21(40.75–94.07) | 3.97(2.52–5.81) |
| Both | 102.01(71.66–138.11) | 3.1(2.18–4.2) | 139.25(97.43–182.01) | 4.24(2.96–5.54) |
| **15 to 19** | | | | |
| Male | 76.43(53.23–105.68) | 4.68(3.26–6.47) | 115.87(82.29–168.44) | 7.09(5.04–10.31) |
| Female | 73.51(52.06–99.63) | 4.35(3.08–5.89) | 134.84(95.4–194.63) | 7.97(5.64–11.51) |
| Both | 149.93(116.35–192.79) | 4.51(3.5–5.8) | 250.71(191.79–354.88) | 7.54(5.77–10.67) |
| **20 to 24** | | | | |
| Male | 64.46(43.52–89.34) | 4.72(3.19–6.55) | 130.02(88.74–192.17) | 9.53(6.5–14.08) |
| Female | 86.32(62.86–111.2) | 5.47(3.98–7.04) | 226.79(164.07–321.92) | 14.36(10.39–20.38) |
| Both | 150.77(119–188.13) | 5.12(4.04–6.39) | 356.8(273.03–495.68) | 12.12(9.27–16.84) |
| **25 to 29** | | | | |
| Male | 52.54(23.42–76.01) | 4.82(2.15–6.98) | 126.97(62.62–182.27) | 11.65(5.75–16.73) |
| Female | 110.18(63.3–144.84) | 7.9(4.54–10.39) | 371.72(216.59–524.07) | 26.66(15.54–37.59) |
| Both | 162.71(92.09–207.11) | 6.55(3.71–8.34) | 498.69(300.84–672.01) | 20.08(12.11–27.06) |
| **30 to 34** | | | | |
| Male | 64.43(8.3–98.61) | 7.09(0.91–10.85) | 137.4(30.05–199.81) | 15.11(3.31–21.98) |
| Female | 176.12(77.97–237.14) | 14.59(6.46–19.64) | 564.39(241.05–792.46) | 46.75(19.97–65.64) |
| Both | 240.55(96.16–318.64) | 11.37(4.54–15.06) | 701.8(290.74–959.12) | 33.16(13.74–45.32) |
| **35 to 39** | | | | |
| Male | 102.46(12.63–160.17) | 12.71(1.57–19.87) | 181.93(34–271.84) | 22.57(4.22–33.73) |
| Female | 298.75(187.34–404.82) | 28.74(18.02–38.95) | 786.74(506.01–1107.9) | 75.69(48.69–106.59) |
| Both | 401.21(217.58–526.79) | 21.74(11.79–28.55) | 968.67(551.88–1295.7) | 52.49(29.91–70.21) |
| **40 to 44** | | | | |
| Male | 199.41(80.29–281.42) | 27.54(11.09–38.87) | 285.69(111.01–408.51) | 39.46(15.33–56.42) |
| Female | 483.66(345.2–644.67) | 55.11(39.33–73.45) | 1037(724.67–1428.26) | 118.15(82.57–162.73) |
| Both | 683.07(464.71–871.47) | 42.65(29.01–54.41) | 1322.69(898.66–1735.79) | 82.58(56.11–108.37) |
| **45 to 49** | | | | |
| Male | 373.55(220.37–505.49) | 57.29(33.8–77.52) | 489.71(289.79–672.78) | 75.1(44.44–103.18) |
| Female | 674.05(498.24–871.89) | 92.2(68.15–119.26) | 1216.13(889.88–1609.61) | 166.34(121.72–220.17) |

*(Continued)*

**Table 2.** (Continued)

| Age group | Total deaths (95%UI) | Age-specific death rates, Number in 100,000 (95%UI) | Total new cases, (95%UI) | Age-specific incidence rates, Number in 100,000 (95%UI) |
|---|---|---|---|---|
| Both | 1047.6(777.62–1281.92) | 75.74(56.22–92.68) | 1705.84(1287.06–2138.69) | 123.33(93.05–154.63) |
| **50 to 54** | | | | |
| Male | 641.77(460.32–852.5) | 111.43(79.93–148.02) | 796.51(576.64–1053.77) | 138.3(100.12–182.97) |
| Female | 856.39(659.3–1110.91) | 141.34(108.82–183.35) | 1319.94(995.1–1728.9) | 217.85(164.24–285.35) |
| Both | 1498.15(1212.66–1803.87) | 126.77(102.61–152.64) | 2116.45(1713.17–2576.47) | 179.09(144.96–218.01) |
| **55 to 59** | | | | |
| Male | 984.36(744.51–1233.06) | 199.17(150.64–249.49) | 1161.59(881.72–1441.35) | 235.03(178.4–291.64) |
| Female | 954.97(750.39–1230.23) | 185.72(145.93–239.25) | 1330.61(1039.08–1723.63) | 258.77(202.07–335.2) |
| Both | 1939.34(1626.63–2266.95) | 192.31(161.3–224.8) | 2492.2(2083.8–2964.59) | 247.13(206.64–293.98) |
| **60 to 64** | | | | |
| Male | 1305.04(1002.59–1579.7) | 317.72(244.09–384.59) | 1474.97(1146.64–1778.53) | 359.09(279.16–432.99) |
| Female | 1089.65(858.27–1399.66) | 247.84(195.21–318.35) | 1380(1081.21–1757.28) | 313.88(245.92–399.69) |
| Both | 2394.7(2030.78–2786.79) | 281.59(238.8–327.7) | 2854.97(2413.54–3304.38) | 335.72(283.81–388.56) |
| **65 to 69** | | | | |
| Male | 1479.73(1159.67–1814.74) | 453.77(355.62–556.5) | 1583.38(1242.18–1907.43) | 485.55(380.92–584.92) |
| Female | 1152.63(891.81–1468.91) | 320.52(247.99–408.47) | 1326.31(1028.29–1668.32) | 368.82(285.95–463.92) |
| Both | 2632.35(2211.89–3026.15) | 383.89(322.57–441.32) | 2909.69(2446.92–3349.02) | 424.33(356.85–488.4) |
| **70 to 74** | | | | |
| Male | 1427.98(1139.34–1761.94) | 611.23(487.68–754.18) | 1444.36(1157.52–1758.87) | 618.24(495.46–752.86) |
| Female | 1054.96(808.05–1322.23) | 430.83(330–539.99) | 1095.99(841.4–1375.88) | 447.59(343.62–561.9) |
| Both | 2482.94(2114.46–2847.83) | 518.91(441.9–595.17) | 2540.35(2163.51–2904.69) | 530.91(452.16–607.06) |
| **75 to 79** | | | | |
| Male | 1135.32(902.54–1381.8) | 788.84(627.1–960.1) | 1062.12(850.16–1273.44) | 737.98(590.71–884.81) |
| Female | 809.6(629.08–994.6) | 528.6(410.74–649.4) | 766.51(597.41–946.8) | 500.47(390.06–618.18) |
| Both | 1944.92(1664.62–2211.55) | 654.68(560.33–744.43) | 1828.63(1552.46–2075.27) | 615.53(522.57–698.55) |
| **80 plus** | | | | |
| Male | 1216.56(1002.95–1420.79) | 1113.54(918.02–1300.48) | 976.2(806.96–1128.24) | 893.53(738.62–1032.7) |
| Female | 1064.43(856.39–1281.38) | 749.56(603.06–902.34) | 860.16(686.21–1048.26) | 605.72(483.22–738.17) |
| Both | 2280.99(1991.68–2537.21) | 907.83(792.68–1009.8) | 1836.36(1600.58–2061.78) | 730.86(637.02–820.58) |

in males was 91.2 (73.9–108.66) in 1990 and 94.78 (77.14–111.80) in 2017 and among females, it was 95.36 (76.16–121.34) in 1990 and 79.31 (64.30–96.63) in 2017 (Table 4).

Among all forms of cancer, the major contributors for mortality in both sexes are lung cancer (12.7%), stomach cancer (9.1%), breast cancer (8.4%), colorectal cancer (6.9%) and esophageal cancer (6%) in 2017. Overall, in both sexes, there has been a significant increase in

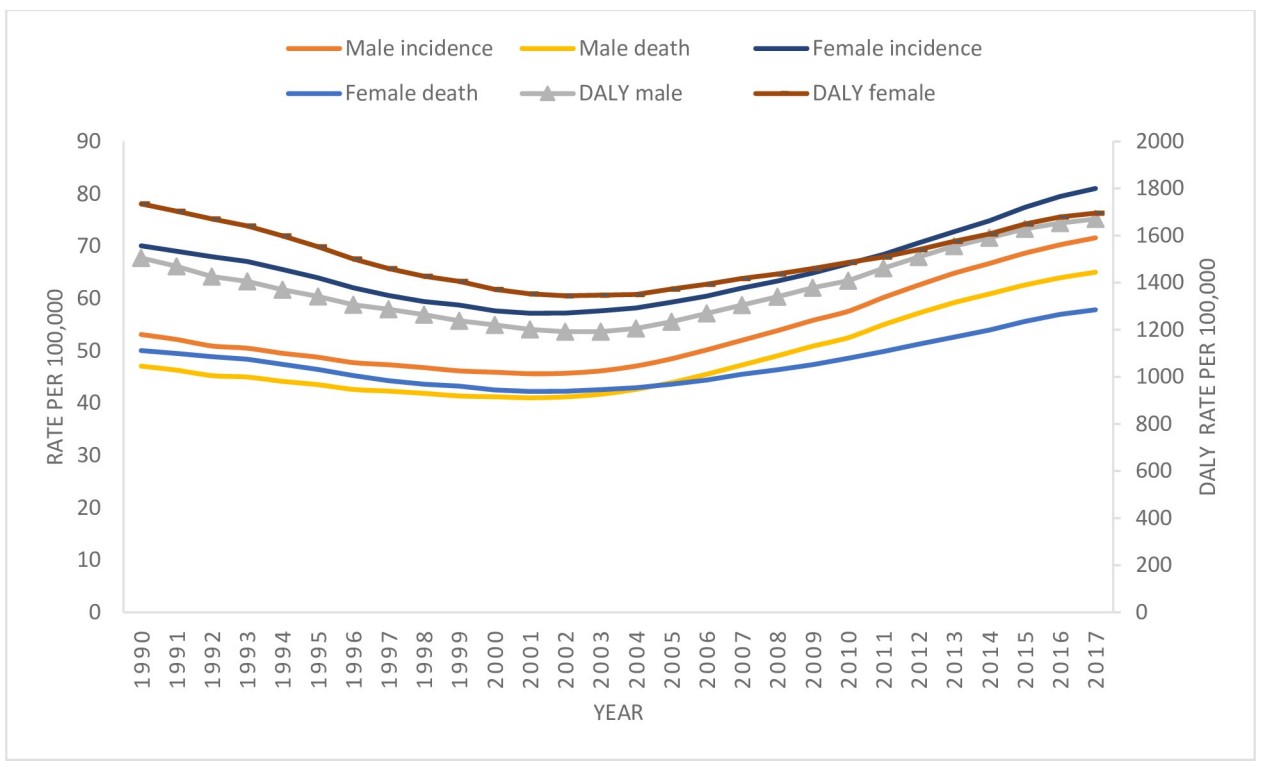

**Fig 1. Trend of cancer by crude incidence rate, mortality rate and DALY in Nepal, 1990–2017.**

mortality from pancreatic cancer, liver cancer, ovarian cancer, kidney cancer, and multiple myeloma with a percentage change of 80.42, 54.30, 51.67, 49.84, and 31.28 respectively from 1990 to 2017.

The five most common causes of death from cancer among males were lung cancer (15%), stomach cancer (10.1%), esophageal cancer (7.8%), lip and oral cavity cancer (7.1%) and prostate cancer (7.1%). Prostate and esophageal cancers have an increasing trend of mortality with percentage change of 27.38 and 8.38 respectively from 1990 to 2017. While, a decreasing trend of mortality was found with a percentage change of 3.70, 28.03, and 16.68. for lung, stomach, and oral cavity cancer from 1990 to 2017.

Among females, the five most common causes of death from cancer were breast cancer (16.6%), lung cancer (10.4%), cervical cancer (10.1%), stomach cancer (8.1%), and colorectal cancer (7.0%). Breast, lung, and colorectal cancer have an increasing trend of incidence with a percentage change of 3.22, 2.61, and 2.20, respectively from 1990 to 2017. Table 2 shows the number of deaths is highest among people of age-group 65–69 years. The age-specific mortality rate was highest among people aged 80 years and older for both males 1113.54 (918.02–1300.48) and females 749.56 (603.06–902.34) per 100,000 population in 2017 (Table 4). Fig 1 shows the increase in the death rate per 100,000 population from 50.05 (39.89–64.21) to 57.84 (46.58–70.64) in females and 47.06 (38.25–56.18) to 65.02 (52.21–77.51) in males from 1990 to 2017.

## Disability-adjusted life years

Cancer accounted for 5.6% of total DALYs in both sexes in 2017 mostly attributed to lung cancer followed by breast and stomach cancer. DALYs (rates per 100,000 population) for all forms

**Table 3. All-age prevalence and age-standardized prevalence rates for different types of cancer and their percentage change by gender in Nepal, 1990–2017.**

| Morphology | All-Age Prevalence (95% UI) | | | Age-Standardized Prevalence Rate (95% UI), per 100 000 | | |
|---|---|---|---|---|---|---|
| | 1990 | 2017 | Change, % | 1990 | 2017 | Change, % |
| **All Neoplasms** | | | | | | |
| Male | 7380.16(5919.44–9128.73) | 16705.85(13142.65–19857.28) | 126.36 | 139.21(112.75–167.82) | 163.13(129.67–192.57) | 17.18 |
| Female | 21496.67(16251.85–28697.23) | 41864.14(31997.05–57261.36) | 94.75 | 350.94(273.07–455.61) | 328.98(254.92–447.38) | -6.26 |
| Both | 28876.83(23258.27–36556.61) | 58569.99(48035.28–74447.91) | 102.83 | 243.75(203.7–299.21) | 252.12(209.89–318.24) | 3.43 |
| **Tracheal, bronchus, and lung cancer** | | | | | | |
| Male | 711.08(467.7–1178.39) | 1298.06(949.87–2081.11) | 82.55 | 13.78(9.03–23.05) | 12.63(9.3–20.21) | -8.39 |
| Female | 419.45(128.93–713.09) | 1076.97(413.86–1764.65) | 156.75 | 8.95(2.56–15.43) | 9.44(3.59–15.48) | 5.51 |
| Both | 1130.54(674.64–1699.5) | 2375.03(1507.12–3264.71) | 110.08 | 11.44(6.71–17.19) | 10.97(6.96–14.95) | -4.14 |
| **Breast cancer** | | | | | | |
| Male | 79.67(61.4–101.74) | 221.29(162.52–283.34) | 177.75 | 1.46(1.14–1.84) | 2.09(1.56–2.64) | 42.87 |
| Female | 5631.11(3722.72–8965.9) | 18652.25(13585.93–33279.31) | 231.24 | 96.77(65.89–150.89) | 146.11(107.11–259.23) | 50.98 |
| Both | 5710.78(3791.46–9040.32) | 18873.53(13800.29–33548.2) | 230.49 | 48.15(32.97–74.61) | 78.24(57.58–138.49) | 62.49 |
| **Cervical cancer** | | | | | | |
| Female | 10595.96(6323.05–14680.8) | 11657.9(7707.73–16734.69) | 10.02 | 160.03(93.92–217.77) | 84.23(56.4–118.08) | -47.37 |
| Both | 10595.96(6323.05–14680.8) | 11657.9(7707.73–16734.69) | 10.02 | 79.17(46.6–107.76) | 45.32(30.18–63.83) | -42.76 |
| **Colon and rectum cancer** | | | | | | |
| Male | 846.58(516.17–1334.13) | 2206.66(1514.49–3307.88) | 160.66 | 16.45(10.07–25.7) | 21.6(14.96–32.31) | 31.3 |
| Female | 889.4(532.46–1555.88) | 2373.52(1797.8–3072.45) | 166.87 | 18.3(11.1–31.6) | 20.99(16.11–27.21) | 14.67 |
| Both | 1735.98(1177.89–2688.47) | 4580.18(3655.55–5854.15) | 163.84 | 17.38(12.04–26.5) | 21.33(17.22–27.24) | 22.74 |
| **Esophageal cancer** | | | | | | |
| Male | 512.54(402.21–646.17) | 978.33(751.99–1212.46) | 90.88 | 9.08(7.2–11.37) | 9.02(7–11.03) | -0.69 |
| Female | 407.29(318.33–531.21) | 506.01(387.5–673.24) | 24.24 | 7.69(5.98–10.06) | 4.24(3.27–5.61) | -44.89 |
| Both | 919.83(772.84–1104.61) | 1484.34(1203.26–1776.97) | 61.37 | 8.42(7.11–10.16) | 6.53(5.34–7.79) | -22.5 |
| **Lip and oral cavity cancer** | | | | | | |
| Male | 2014.87(1319.66–3058.99) | 3239.1(2269.02–4292.24) | 60.76 | 35.33(23.46–53.1) | 29.84(21.03–39.24) | -15.54 |
| Female | 1061.65(785.69–1387.42) | 2686.19(2048.32–3362.25) | 153.02 | 20.05(14.91–26.46) | 21.89(16.93–27.38) | 9.16 |
| Both | 3076.52(2263.87–4197.14) | 5925.29(4702.84–7195.29) | 92.6 | 27.92(20.64–37.66) | 25.76(20.56–31.13) | -7.73 |
| **Stomach cancer** | | | | | | |
| Male | 1020.92(784.06–1300.04) | 1392.52(1047.41–1758.7) | 36.4 | 19.69(15.07–24.99) | 13.6(10.35–17.06) | -30.92 |
| Female | 742.29(560.67–1064.87) | 1136(887.03–1457.83) | 53.04 | 14.09(10.76–20.06) | 9.72(7.64–12.49) | -30.98 |
| Both | 1763.21(1456.36–2147.1) | 2528.52(2112.66–3012.13) | 43.4 | 16.98(14.07–20.51) | 11.59(9.72–13.76) | -31.76 |
| **Bladder cancer** | | | | | | |
| Male | 419.43(302.94–715.77) | 1161.38(832.9–2062.62) | 176.9 | 8.77(6.37–14.55) | 11.64(8.38–20.31) | 32.8 |
| Female | 209.27(138.14–315.99) | 529(370.92–690.68) | 152.79 | 4.42(2.95–6.52) | 4.69(3.31–6.12) | 6.06 |
| Both | 628.69(476.9–929.26) | 1690.38(1300.65–2577.82) | 168.87 | 6.62(5.03–9.54) | 7.97(6.2–11.97) | 20.37 |
| **Brain and nervous system cancer** | | | | | | |
| Male | 442.93(240.07–787.31) | 675.61(321.81–1312.34) | 52.53 | 5.03(2.44–9.93) | 5.43(2.61–10.44) | 8.05 |
| Female | 732.83(199.59–1402.71) | 634.66(415.51–985.36) | -13.4 | 7.48(2.45–13.87) | 4.64(3.08–7.13) | -37.96 |
| Both | 1175.76(627.46–1882.71) | 1310.27(855.02–2187.12) | 11.44 | 6.23(3.99–9.83) | 5.03(3.31–8.27) | -19.37 |
| **Gallbladder and biliary tract cancer** | | | | | | |
| Male | 59.84(41.69–90.72) | 141.76(86.53–200.36) | 136.88 | 1.21(0.86–1.84) | 1.41(0.86–1.97) | 17.09 |
| Female | 112.85(77.09–225.88) | 280.98(197.51–433.79) | 148.98 | 2.38(1.64–4.88) | 2.48(1.73–3.84) | 3.88 |
| Both | 172.69(127.84–297.86) | 422.73(314.93–568.48) | 144.79 | 1.78(1.33–3.08) | 1.97(1.48–2.67) | 10.65 |
| **Hodgkin lymphoma** | | | | | | |
| Male | 414.17(270.01–684.3) | 271.2(153.32–541.67) | -34.52 | 5.22(3.45–8.76) | 2.15(1.22–4.32) | -58.74 |
| Female | 181.41(119.43–261.99) | 176.89(101.65–331.07) | -2.49 | 2.24(1.54–3.19) | 1.18(0.69–2.17) | -47.45 |

*(Continued)*

**Table 3.** (Continued)

| Morphology | All-Age Prevalence (95% UI) | | | Age-Standardized Prevalence Rate (95% UI), per 100 000 | | |
|---|---|---|---|---|---|---|
| | 1990 | 2017 | Change, % | 1990 | 2017 | Change, % |
| Both | 595.58(412.73–898.71) | 448.09(268.73–823.58) | -24.76 | 3.75(2.68–5.62) | 1.65(1.01–3.03) | -56.04 |
| **Kidney cancer** | | | | | | |
| Male | 469.5(264–892.54) | 1296.9(779.05–2204.28) | 176.23 | 6.15(3.58–10.28) | 11.11(6.74–18.9) | 80.63 |
| Female | 573.03(294.48–1109.54) | 1105.95(700.09–1699.42) | 93 | 6.9(4.01–12.23) | 8.37(5.37–12.8) | 21.31 |
| Both | 1042.53(613.04–1765.67) | 2402.85(1582.92–3642.14) | 130.48 | 6.52(4.02–9.89) | 9.71(6.45–14.67) | 48.86 |
| **Larynx cancer** | | | | | | |
| Male | 1255.96(959.11–1629.77) | 2027.69(1537.36–2674.78) | 61.45 | 22.98(17.76–29.36) | 18.73(14.34–24.66) | -18.49 |
| Female | 561.11(391.95–744.92) | 842.35(583.53–1097.08) | 50.12 | 10.37(7–13.75) | 6.93(4.8–8.98) | -33.15 |
| Both | 1817.07(1458.61–2208.4) | 2870.05(2326.55–3452.05) | 57.95 | 16.87(13.56–20.56) | 12.58(10.25–15.05) | -25.39 |
| **Leukemia** | | | | | | |
| Male | 1141.86(713.26–1762.83) | 1247.41(814.99–1695.2) | 9.24 | 12.23(8.73–16.49) | 9.93(6.49–13.37) | -18.8 |
| Female | 1333.37(705.47–3013.68) | 1393.71(1000.75–1884.26) | 4.53 | 14.07(8.66–27.14) | 10(7.28–13.35) | -28.98 |
| Both | 2475.23(1508.13–4576.06) | 2641.12(1999.85–3326.07) | 6.7 | 13.14(9.34–20.46) | 9.97(7.72–12.42) | -24.09 |
| **Liver cancer** | | | | | | |
| Male | - | - | - | - | - | - |
| Female | - | - | - | - | - | - |
| Both | - | - | - | - | - | - |
| **Malignant skin melanoma** | | | | | | |
| Male | 54.18(34.43–99.95) | 111.21(71.66–185.26) | 105.27 | 0.9(0.59–1.6) | 1.01(0.65–1.67) | 11.87 |
| Female | 41.68(20.83–106.61) | 123.11(73.1–261.95) | 195.37 | 0.74(0.38–1.82) | 0.97(0.58–2.08) | 30.8 |
| Both | 95.86(62.76–182.97) | 234.33(155.25–404.86) | 144.44 | 0.82(0.55–1.54) | 0.99(0.67–1.71) | 20.6 |
| **Mesothelioma** | | | | | | |
| Male | 25.51(12.66–51.6) | 66.26(42.52–98.84) | 159.78 | 0.47(0.24–0.94) | 0.63(0.41–0.92) | 33.79 |
| Female | 21.37(8.29–43.11) | 29.03(17.66–45.27) | 35.9 | 0.36(0.15–0.71) | 0.23(0.14–0.35) | -36.87 |
| Both | 46.87(24.56–90.87) | 95.29(64.69–137.44) | 103.31 | 0.42(0.22–0.8) | 0.42(0.29–0.6) | 0.29 |
| **Multiple myeloma** | | | | | | |
| Male | 40.86(23.98–68.49) | 127.75(77.76–223.1) | 212.68 | 0.74(0.45–1.27) | 1.17(0.72–2.03) | 57.61 |
| Female | 43.49(28.71–68.06) | 159.48(104.78–255.04) | 266.71 | 0.88(0.58–1.41) | 1.35(0.89–2.15) | 53.89 |
| Both | 84.34(58.39–131.98) | 287.23(199.4–439.3) | 240.54 | 0.81(0.58–1.26) | 1.27(0.89–1.92) | 56.4 |
| **Nasopharynx cancer** | | | | | | |
| Male | 351.87(232.52–542.22) | 585.54(404.14–833.89) | 66.41 | 5.56(3.88–8.06) | 5.07(3.55–6.95) | -8.77 |
| Female | 307.78(181.01–532.63) | 358.38(220.35–571.07) | 16.44 | 4.54(2.96–7.08) | 2.62(1.71–3.98) | -42.3 |
| Both | 659.65(470.32–936.44) | 943.93(679.7–1260.67) | 43.09 | 5.07(3.81–6.77) | 3.79(2.8–4.93) | -25.33 |
| **Non-Hodgkin lymphoma** | | | | | | |
| Male | 405.89(264.35–648.81) | 818.4(556.63–1220.02) | 101.63 | 6.01(3.81–8.89) | 7.24(4.93–10.7) | 20.43 |
| Female | 254.7(171.92–364.53) | 634.76(378.92–917.83) | 149.21 | 3.93(2.72–5.37) | 5.09(3.07–7.36) | 29.63 |
| Both | 660.59(492.54–890.03) | 1453.16(1093.63–1855.61) | 119.98 | 4.98(3.93–6.29) | 6.12(4.62–7.78) | 22.93 |
| **Non-melanoma skin cancer** | | | | | | |
| Male | - | - | - | - | - | - |
| Female | - | - | - | - | - | - |
| Both | - | - | - | - | - | - |
| **Other pharynx cancer** | | | | | | |
| Male | 488.97(281.92–773.63) | 1217.26(821.12–1728.25) | 148.94 | 8.62(4.95–13.69) | 11.12(7.56–15.72) | 29.01 |
| Female | 294.87(218.23–403.76) | 674.67(508.65–947.84) | 128.8 | 5.41(4–7.42) | 5.51(4.14–7.73) | 1.96 |
| Both | 783.85(578.35–1098.48) | 1891.93(1452.28–2479.3) | 141.36 | 7.07(5.18–9.93) | 8.21(6.33–10.69) | 16.07 |
| **Ovarian cancer** | | | | | | |

*(Continued)*

**Table 3.** (Continued)

| Morphology | All-Age Prevalence (95% UI) | | | Age-Standardized Prevalence Rate (95% UI), per 100 000 | | |
| --- | --- | --- | --- | --- | --- | --- |
| | **1990** | **2017** | **Change, %** | **1990** | **2017** | **Change, %** |
| Female | 583.13(389.39–1067.44) | 2193.04(1566.37–2905.41) | 276.08 | 9.5(6.4–16.97) | 16.49(11.97–21.95) | 73.62 |
| Both | 583.13(389.39–1067.44) | 2193.04(1566.37–2905.41) | 276.08 | 4.68(3.15–8.35) | 8.76(6.34–11.68) | 87.19 |
| **Pancreatic cancer** | | | | | | |
| Male | 61.83(39.7–99.14) | 210.39(135.27–337.71) | 240.28 | 1.19(0.76–1.89) | 2.07(1.32–3.29) | 74.12 |
| Female | 41.4(27.91–56.17) | 173.65(115.8–232.19) | 319.44 | 0.9(0.58–1.22) | 1.56(1.04–2.08) | 73.83 |
| Both | 103.23(71.39–145.43) | 384.04(266.69–541.52) | 272.02 | 1.05(0.71–1.47) | 1.8(1.26–2.52) | 72.22 |
| **Prostate cancer** | | | | | | |
| Male | 753.21(543.74–1023.05) | 3339.31(2384.22–4472.59) | 343.34 | 19.32(13.61–26.33) | 34.89(24.59–46.49) | 80.6 |
| Both | 753.21(543.74–1023.05) | 3339.31(2384.22–4472.59) | 343.34 | 9.73(6.85–13.28) | 16.41(11.58–21.93) | 68.71 |
| **Testicular cancer** | | | | | | |
| Male | 261.98(160.96–369.78) | 230.97(131.3–339.09) | -11.84 | 3.27(2.04–4.62) | 1.72(0.94–2.51) | -47.34 |
| Both | 261.98(160.96–369.78) | 230.97(131.3–339.09) | -11.84 | 1.6(1–2.26) | 0.78(0.44–1.14) | -51.06 |
| **Thyroid cancer** | | | | | | |
| Male | 232.73(161.46–327.78) | 679.12(471.82–943.71) | 191.81 | 3.67(2.6–5.07) | 5.86(4.16–8.14) | 59.56 |
| Female | 935.6(587.63–1588.71) | 2902.58(1921.84–4666.46) | 210.24 | 13(8.4–22.27) | 20.02(13.41–31.95) | 53.98 |
| Both | 1168.33(795.44–1815.5) | 3581.71(2539.55–5204.03) | 206.57 | 8.37(5.92–12.82) | 13.52(9.78–19.49) | 61.65 |
| **Uterine cancer** | | | | | | |
| Female | 778.75(445.47–1094.34) | 1734.79(1212.31–2354.92) | 122.77 | 15.04(8.77–20.99) | 14.38(10.16–19.46) | -4.34 |
| Both | 778.75(445.47–1094.34) | 1734.79(1212.31–2354.92) | 122.77 | 7.31(4.27–10.23) | 7.52(5.32–10.2) | 2.85 |

of cancer decreased by 15.96% from 2519.16 (2150.05–2946.53) in 1990 to 2117.11(1816.02–2397.29) in 2017.

In males, lung, stomach, and esophageal cancers claimed most DALYs. In females, breast cancer claimed the highest DALYs, followed by cervix and lung cancers. Table 4 shows all age DALYs and age-standardized DALYs rates. Similar to mortality rate, the DALYs of pancreatic cancer [67.4% increase; 34.37(23.46–48.05) to 57.54(39.95–80.98)], Ovarian cancer [49.32% increase;26.19(17.81–45.78) to 39.1(28.95–51.52)), and Liver cancer (39.8% increase; 55.98 (40.39–74.52) to 78.25(53.82–122.81)] showed the greatest increases, while Testicular cancer (67.62% decrease; 9.45 (5.88–13.2) to 3.06(1.77–4.28)], Hodgkin lymphoma [61.02% decrease; 44.88 (31.49–67.27) to 17.49 (10.5–32.11)] and cervical cancer [53.82% decrease; 269.86 (158.98–357.53) to 124.63(87.09–170.91)] showed the greatest decreases in DALY rates over time (Table 5).

## Risk factors

The leading risk factors associated with the highest DALYs were tobacco 315.2 (248.3–384.3) [smoking 257.1 (199.7–317.1), chewing tobacco 72.1 (53.5–91.0)], diet 126.1 (92.8–159.8), unsafe sex 102.8 (71.5–141.2), alcohol use 96.9 (66.5–129.2), air pollution 74.0 (44.4–109.1), low fruit intake 58.0 (31.3–90.0), and high sodium intake 36.2 (6.6–71.9) (Fig 2).

## Discussion

Nepal is facing fluctuations in cancer prevalence and DALYs over the past 27 years. The top 5 cancers according to the age-standardized incidence rate, both sexes are breast, lung, cervical, stomach and oral cavity cancer. The burden of cancer particularly; cancer-associated mortality rate and disability-adjusted life years due to cancer is increasing. With the increased

**Table 4. All-age deaths and age-standardized mortality rates for different types of cancer and their percentage change by gender in Nepal, 1990–2017.**

| Morphology | All-Age Deaths, (95% UI) | | | Age-Standardized Mortality Rate (95% UI), per 100000 | | |
|---|---|---|---|---|---|---|
| | 1990 | 2017 | Change,% | 1990 | 2017 | Change,% |
| **All Neoplasms** | | | | | | |
| Male | 4568.75(3713.39–5454.4) | 9294.53(7463.08–11079.92) | 103.44 | 91.2(73.9–108.66) | 94.78(77.14–111.8) | 3.92 |
| Female | 4836.97(3855.41–6205.73) | 9020(7265.11–11017.63) | 86.48 | 95.36(76.16–121.34) | 79.31(64.3–96.63) | -16.82 |
| Both | 9405.73(8059.03–10942.5) | 18314.54(16013.37–20596.07) | 94.72 | 93.21(81–107.54) | 86.56(75.59–97.03) | -7.14 |
| **Tracheal, bronchus, and lung cancer** | | | | | | |
| Male | 709.9(466.05–1185.32) | 1391.89(1016.03–2286.62) | 96.07 | 14.67(9.48–24.76) | 14.12(10.45–23.05) | -3.7 |
| Female | 362.76(106.25–623.72) | 936.09(353.67–1539.96) | 158.05 | 8.33(2.28–14.81) | 8.55(3.17–14.04) | 2.61 |
| Both | 1072.66(654.64–1641.03) | 2327.97(1519.9–3236.57) | 117.03 | 11.59(6.9–17.58) | 11.2(7.33–15.55) | -3.41 |
| **Breast cancer** | | | | | | |
| Male | 14.11(10.94–17.97) | 34.88(26.12–43.78) | 147.08 | 0.3(0.23–0.37) | 0.37(0.28–0.45) | 23.48 |
| Female | 648.66(421.56–1084.37) | 1500.59(1088.19–2764.06) | 131.34 | 12.28(8.23–20.77) | 12.68(9.25–23.56) | 3.22 |
| Both | 662.77(434.76–1097.43) | 1535.46(1122.91–2802.91) | 131.67 | 6.17(4.18–10.32) | 6.88(5.07–12.66) | 11.51 |
| **Cervical cancer** | | | | | | |
| Female | 941.86(556.6–1241.73) | 915.38(651.02–1235.5) | -2.81 | 17.16(10.2–22.72) | 7.52(5.43–10.07) | -56.17 |
| Both | 941.86(556.6–1241.73) | 915.38(651.02–1235.5) | -2.81 | 8.4(5–11.12) | 3.98(2.87–5.33) | -52.62 |
| **Colon and rectum cancer** | | | | | | |
| Male | 245.66(148.81–385.8) | 631.18(435.74–942.76) | 156.93 | 5.37(3.26–8.31) | 6.76(4.71–10.06) | 25.9 |
| Female | 255.69(155.27–440.61) | 635.65(487.26–816.67) | 148.6 | 6(3.75–10.06) | 6.13(4.71–7.88) | 2.2 |
| Both | 501.35(345.63–769.52) | 1266.83(1019.83–1631.35) | 152.68 | 5.7(4–8.6) | 6.44(5.23–8.29) | 13.11 |
| **Esophageal cancer** | | | | | | |
| Male | 338.21(268.54–421.02) | 727.79(566.67–888.69) | 115.19 | 6.7(5.37–8.34) | 7.27(5.71–8.76) | 8.38 |
| Female | 274.49(214.06–361.89) | 366.62(282.47–492.28) | 33.57 | 5.96(4.59–8.01) | 3.34(2.56–4.42) | -44 |
| Both | 612.69(517.48–738.72) | 1094.41(897.4–1299.48) | 78.62 | 6.36(5.39–7.73) | 5.2(4.3–6.15) | -18.23 |
| **Lip and oral cavity cancer** | | | | | | |
| Male | 411.33(271.25–616.04) | 663.47(469.6–880.56) | 61.3 | 7.81(5.2–11.46) | 6.5(4.65–8.57) | -16.68 |
| Female | 179.58(133.08–235.25) | 382.99(302.28–482.81) | 113.27 | 3.96(2.97–5.22) | 3.52(2.79–4.39) | -11.08 |
| Both | 590.91(433.28–805.81) | 1046.46(825.94–1271.56) | 77.09 | 5.94(4.39–8) | 4.94(3.96–5.93) | -16.78 |
| **Stomach cancer** | | | | | | |
| Male | 643.61(494.68–818.41) | 938.14(713.57–1175.18) | 45.76 | 13.57(10.43–17.1) | 9.77(7.44–12.2) | -28.03 |
| Female | 458.14(347.3–653.69) | 734.98(581.64–942.56) | 60.43 | 9.63(7.42–13.74) | 6.68(5.29–8.52) | -30.69 |
| Both | 1101.76(915.06–1330.16) | 1673.12(1409.55–1976.95) | 51.86 | 11.65(9.66–13.98) | 8.14(6.88–9.56) | -30.14 |
| **Bladder cancer** | | | | | | |
| Male | 74.06(53.37–121.81) | 192.85(138.66–324.1) | 160.38 | 1.96(1.41–3.12) | 2.28(1.62–3.76) | 16.07 |
| Female | 36.38(24.06–54) | 77.99(55.5–102.21) | 114.35 | 0.96(0.63–1.41) | 0.8(0.57–1.05) | -16.36 |
| Both | 110.45(83.85–158.35) | 270.83(208.14–400.25) | 145.21 | 1.46(1.09–2.07) | 1.48(1.14–2.17) | 1.54 |
| **Brain and nervous system cancer** | | | | | | |
| Male | 123.23(57.44–246.59) | 237.47(116.17–439.64) | 92.71 | 1.72(0.74–3.71) | 2.06(1.02–3.77) | **19.76** |
| Female | 173.97(58.75–322.76) | 214.96(144.64–329.19) | 23.56 | 2.25(0.87–4.31) | 1.66(1.14–2.52) | -26.19 |
| Both | 297.2(194.48–461.99) | 452.44(302.49–738.58) | 52.23 | 1.98(1.44–3.07) | 1.85(1.25–2.98) | -6.48 |
| **Gallbladder and biliary tract cancer** | | | | | | |
| Male | 83.51(59.43–126.71) | 218.52(133.8–306.8) | 161.68 | 1.89(1.38–2.83) | 2.32(1.42–3.22) | 22.7 |
| Female | 157.58(107.77–320.91) | 412.68(288.01–647.34) | 161.88 | 3.71(2.53–7.76) | 3.85(2.67–5.99) | 3.68 |
| Both | 241.09(179.99–413.16) | 631.2(472.87–850.38) | 161.81 | 2.79(2.09–4.85) | 3.13(2.35–4.24) | 12.21 |
| **Hodgkin lymphoma** | | | | | | |
| Male | 107.41(70.68–179.1) | 75.73(43.84–149.1) | -29.49 | 1.53(1.01–2.57) | 0.65(0.38–1.28) | -57.54 |
| Female | 45.49(30.87–64.74) | 43.32(25.71–80.54) | -4.78 | 0.63(0.44–0.9) | 0.31(0.19–0.57) | -50.52 |

*(Continued)*

Table 4. (Continued)

| Morphology | All-Age Deaths, (95% UI) | | | Age-Standardized Mortality Rate (95% UI), per 100000 | | |
| --- | --- | --- | --- | --- | --- | --- |
| | 1990 | 2017 | Change,% | 1990 | 2017 | Change,% |
| Both | 152.89(108.39–230.27) | 119.05(73.08–213.35) | -22.14 | 1.09(0.78–1.64) | 0.47(0.3–0.85) | -56.46 |
| **Kidney cancer** | | | | | | |
| Male | 24.38(14.57–38.4) | 87.54(55.08–145) | 259.04 | 0.46(0.27–0.7) | 0.87(0.56–1.43) | 87.36 |
| Female | 18.9(10.94–31.41) | 44.8(29.73–66.3) | 137.09 | 0.36(0.2–0.57) | 0.39(0.26–0.58) | 9.29 |
| Both | 43.28(26.97–61.51) | 132.34(90.04–197.85) | 205.79 | 0.41(0.25–0.56) | 0.62(0.42–0.92) | 49.84 |
| **Larynx cancer** | | | | | | |
| Male | 319.1(243.12–412.85) | 488.69(373.26–642.16) | 53.14 | 6.22(4.78–8.13) | 4.72(3.63–6.15) | -24.09 |
| Female | 142.22(95.01–190.27) | 187.26(127.14–243.28) | 31.67 | 2.84(1.83–3.78) | 1.61(1.08–2.1) | -43.2 |
| Both | 461.32(369.19–565.98) | 675.94(547.61–818.32) | 46.52 | 4.57(3.68–5.61) | 3.09(2.55–3.71) | -32.41 |
| **Leukemia** | | | | | | |
| Male | 259.69(188.4–336.58) | 414.04(280.19–556.81) | 59.43 | 3.77(2.63–5.08) | 3.77(2.55–4.99) | -0.11 |
| Female | 264.14(167.71–472.96) | 389.93(287.71–509.55) | 47.63 | 3.79(2.66–5.92) | 3.08(2.28–4.02) | -18.9 |
| Both | 523.83(382.11–756.57) | 803.97(633.99–1002.09) | 53.48 | 3.78(3.03–4.7) | 3.4(2.71–4.2) | -10 |
| **Liver cancer** | | | | | | |
| Male | 134.84(77.59–194.08) | 494.98(282.41–841.58) | 267.08 | 2.66(1.51–3.75) | 4.89(2.8–8.31) | 84.11 |
| Female | 80.26(42.82–115.32) | 226.18(159.51–315.02) | 181.82 | 1.74(0.96–2.48) | 2.08(1.47–2.87) | 19.25 |
| Both | 215.1(154.76–286.7) | 721.15(500.31–1119.53) | 235.27 | 2.22(1.6–2.93) | 3.42(2.38–5.27) | 54.3 |
| **Malignant skin melanoma** | | | | | | |
| Male | 11.03(7.13–19.79) | 21.76(14.32–35.54) | 97.28 | 0.21(0.14–0.38) | 0.22(0.15–0.36) | 3.67 |
| Female | 8.33(4.28–20.55) | 20.4(12.39–44.07) | 144.91 | 0.18(0.1–0.43) | 0.19(0.11–0.4) | 2.53 |
| Both | 19.36(12.9–36.07) | 42.16(28.47–73.26) | 117.78 | 0.2(0.13–0.36) | 0.2(0.14–0.36) | 2.62 |
| **Mesothelioma** | | | | | | |
| Male | 8.33(4.34–16.49) | 23.1(15–32.85) | 177.33 | 0.17(0.09–0.33) | 0.23(0.15–0.33) | 39.32 |
| Female | 7.92(3.2–15.38) | 11.77(7.63–17.83) | 48.65 | 0.15(0.06–0.29) | 0.1(0.07–0.15) | -34.58 |
| Both | 16.24(8.57–30.88) | 34.87(24.23–49.08) | 114.63 | 0.16(0.09–0.3) | 0.16(0.11–0.23) | 1.56 |
| **Multiple myeloma** | | | | | | |
| Male | 32.84(19.53–56.82) | 92.06(56.45–154.89) | 180.34 | 0.67(0.41–1.17) | 0.92(0.57–1.52) | 36.55 |
| Female | 36.3(24.12–58.07) | 113.27(75.45–177.84) | 212.02 | 0.84(0.56–1.37) | 1.05(0.7–1.65) | 25.76 |
| Both | 69.14(49.09–107.24) | 205.33(146.78–307.19) | 196.97 | 0.76(0.54–1.18) | 0.99(0.71–1.49) | 31.28 |
| **Nasopharynx cancer** | | | | | | |
| Male | 67.43(48.47–89.26) | 117.62(88.01–150.36) | 74.44 | 1.23(0.91–1.6) | 1.12(0.85–1.41) | -8.98 |
| Female | 50.91(38.06–66.41) | 61.77(46.94–78.12) | 21.34 | 0.91(0.69–1.18) | 0.51(0.39–0.64) | -44.12 |
| Both | 118.34(96.25–141.9) | 179.39(148.16–212.43) | 51.6 | 1.07(0.88–1.28) | 0.8(0.66–0.94) | -25.64 |
| **Non-Hodgkin lymphoma** | | | | | | |
| Male | 112.27(71.35–167.69) | 251.33(172.3–370.16) | 123.87 | 1.96(1.21–2.75) | 2.43(1.68–3.54) | 24.34 |
| Female | 69.72(48.22–95.94) | 185.17(111.39–266.08) | 165.59 | 1.27(0.86–1.69) | 1.6(0.96–2.29) | 25.7 |
| Both | 181.99(143.41–233.33) | 436.5(333.01–556.26) | 139.85 | 1.61(1.26–1.98) | 1.99(1.52–2.52) | 23.22 |
| **Non-melanoma skin cancer** | | | | | | |
| Male | 52.62(30.9–71.12) | 123.24(78.02–156.15) | 134.21 | 1.34(0.77–1.84) | 1.46(0.92–1.83) | 8.8 |
| Female | 11.11(7.3–18.18) | 26.65(20.7–34.19) | 139.79 | 0.31(0.21–0.48) | 0.29(0.23–0.37) | -6.48 |
| Both | 63.74(40.78–85.15) | 149.9(100.87–185.3) | 135.18 | 0.82(0.52–1.09) | 0.83(0.56–1.01) | 0.24 |
| **Other pharynx cancer** | | | | | | |
| Male | 209.57(120.29–333.28) | 511.33(348.07–715.73) | 143.99 | 3.92(2.24–6.25) | 4.89(3.35–6.82) | 24.98 |
| Female | 126.03(93.54–172.3) | 263.77(199.11–374.64) | 109.29 | 2.5(1.83–3.45) | 2.28(1.73–3.24) | -8.95 |
| Both | 335.6(245.74–470.46) | 775.1(593.9–1013.75) | 130.96 | 3.23(2.37–4.55) | 3.53(2.71–4.6) | 9.08 |
| **Ovarian cancer** | | | | | | |

(Continued)

**Table 4.** (Continued)

| Morphology | All-Age Deaths, (95% UI) | | | Age-Standardized Mortality Rate (95% UI), per 100000 | | |
|---|---|---|---|---|---|---|
| | **1990** | **2017** | **Change,%** | **1990** | **2017** | **Change,%** |
| Female | 95.39(65.35–164.45) | 314.65(233.22–416.45) | 229.84 | 1.91(1.31–3.23) | 2.69(2–3.55) | 40.99 |
| Both | 95.39(65.35–164.45) | 314.65(233.22–416.45) | 229.84 | 0.94(0.64–1.58) | 1.42(1.05–1.87) | 51.67 |
| **Pancreatic cancer** | | | | | | |
| Male | 77.8(49.71–123.98) | 303.9(196.88–480.77) | 290.59 | 1.68(1.07–2.67) | 3.18(2.08–4.97) | 88.69 |
| Female | 58.36(37.58–80.14) | 259.53(171.29–349.8) | 344.69 | 1.41(0.88–1.95) | 2.47(1.64–3.28) | 74.41 |
| Both | 136.17(92.53–190.73) | 563.42(391.9–785.15) | 313.78 | 1.56(1.04–2.16) | 2.81(1.95–3.9) | 80.42 |
| **Prostate cancer** | | | | | | |
| Male | 216.48(152.68–294.76) | 659.63(469.83–866.99) | 204.71 | 6.57(4.51–8.95) | 8.37(5.99–10.81) | 27.38 |
| Both | 216.48(152.68–294.76) | 659.63(469.83–866.99) | 204.71 | 3.24(2.24–4.42) | 3.8(2.71–4.92) | 17.23 |
| **Testicular cancer** | | | | | | |
| Male | 29.71(18.6–41.64) | 18.69(11.49–25.53) | -37.08 | 0.39(0.25–0.55) | 0.15(0.1–0.21) | -61.26 |
| Both | 29.71(18.6–41.64) | 18.69(11.49–25.53) | -37.08 | 0.19(0.13–0.27) | 0.07(0.04–0.09) | -63.97 |
| **Thyroid cancer** | | | | | | |
| Male | 20.11(14.11–27.28) | 48.51(35.84–64.01) | 141.19 | 0.39(0.27–0.52) | 0.48(0.36–0.63) | 23.58 |
| Female | 40.57(27.14–70.89) | 78.11(55.76–127.31) | 92.52 | 0.77(0.52–1.39) | 0.67(0.48–1.08) | -13.24 |
| Both | 60.69(46.17–88.64) | 126.62(102.27–166.48) | 108.65 | 0.58(0.45–0.85) | 0.58(0.47–0.77) | 0.43 |
| **Uterine cancer** | | | | | | |
| Female | 86.97(51.53–121.05) | 140.72(101.25–187.95) | 61.8 | 1.91(1.18–2.65) | 1.26(0.91–1.68) | -34.12 |
| Both | 86.97(51.53–121.05) | 140.72(101.25–187.95) | 61.8 | 0.93(0.58–1.3) | 0.66(0.48–0.88) | -29.2 |

availability of advanced and sensitive diagnostic modalities for early detection, the incidence of asymptomatic cancers with an indolent course like pancreatic, ovarian, liver, kidney and thyroid cancer is increasing over 27 years in both sexes while the incidence of symptomatic but hidden cancers like larynx cancer, esophageal cancer, nasopharynx cancer, Hodgkin lymphoma, and cervical cancer is decreasing. Although the incidence of cancer was seen highest in the age group 60–70 years in both sexes, the age-specific incidence rate is highest among people aged 80 years because increasing age is a major risk factor for cancer due to decreasing telomerase activity, increasing exposure, and a slower rate of cell development [19]. This corroborates with global data as 60% of cancers occur in people 65 years of age or older worldwide. The elderly population often miss out on health education and awareness campaigns; hence strategies should be formulated to address this population while planning and conducting public health campaigns.

## Lung cancer

Lung cancer ranked the top in terms of incidence and prevalence and also had the highest death rate. It is the most common cancer in males and the second most common cancer in females according to ASIR. In women, lung cancer has an increasing trend of incidence, but a decreasing trend was observed in males with a male-female ratio of 1.625. This increase in trend due to advanced and sensitive diagnosis with low dose-CT leading to early detection. However, there has been significant development in the radiotherapy and chemotherapy modalities, but ongoing risk factors continue to be a threat.

The most important risk factor for lung cancer is smoking. Most smokers start at a young age and a cumulative exposure to smoking over the years leads to multiple cancers [20]. Like many South Asian countries, Nepal is an agricultural country that produces tobacco as a cash crop and has a population that indulges in chewing tobacco, using hookah or pipes for

**Table 5. All-age disability-adjusted life-years (DALYs) and age-standardized DALY rates for different types of cancer and their percentage change by gender in Nepal, 1990–2017.**

| Morphology | All-Age DALYs (95% UI) | | | Age-standardized DALY Rate (95% UI), per 100 000 | | |
|---|---|---|---|---|---|---|
| | 1990 | 2017 | Change, % | 1990 | 2017 | Change, % |
| **Neoplasms** | | | | | | |
| Male | 146175.39(116397.81–175572.63) | 238845.82(188336.68–288138.93) | 63.4 | 2350.15(1907.21–2805.11) | 2143.45(1685.08–2571.86) | -8.8 |
| Female | 167688.42(131853.89–220727.37) | 264566.75(209338.85–325872.84) | 57.77 | 2690.41(2141.35–3463.71) | 2081.79(1673.56–2566.46) | -22.62 |
| Both | 313863.8(259964.88–376772.9) | 503412.57(426765.17–571694.75) | 60.39 | 2519.16(2150.05–2946.53) | 2117.11(1816.02–2397.29) | -15.96 |
| **Tracheal, bronchus, and lung cancer** | | | | | | |
| Male | 19275.04(12737.88–31709.67) | 31886.51(22952.45–51322.62) | 65.43 | 350.93(231.03–581.03) | 297.26(215.83–479.68) | -15.29 |
| Female | 9508.16(3085.33–15730.92) | 22874.21(8995.43–37089.37) | 140.57 | 187.68(56.63–317.71) | 191.24(75–311.39) | 1.9 |
| Both | 28783.2(17898.09–43999.81) | 54760.72(35829.35–76709.73) | 90.25 | 271.97(165.75–415.8) | 242.32(158.18–338.39) | -10.9 |
| **Breast cancer** | | | | | | |
| Male | 422.89(322.06–542.59) | 879.35(645.38–1130.78) | 107.94 | 7.4(5.69–9.42) | 8.16(6.04–10.37) | 10.28 |
| Female | 22324.2(14056.99–37099.87) | 47595.35(34266.87–87107.37) | 113.2 | 374.97(241.33–618.64) | 372.56(268.74–678.96) | -0.64 |
| Both | 22747.09(14418.07–37548.15) | 48474.7(35210.61–88091.39) | 113.1 | 187.1(121.59–305.93) | 200.55(145.63–362.3) | 7.18 |
| **Cervical cancer** | | | | | | |
| Female | 33751.39(19715.39–45256.96) | 30737.74(21361.55–42196.93) | -8.93 | 551.11(324.7–728.41) | 235.31(164.72–322.04) | -57.3 |
| Both | 33751.39(19715.39–45256.96) | 30737.74(21361.55–42196.93) | -8.93 | 269.86(158.98–357.53) | 124.63(87.09–170.91) | -53.82 |
| **Colon and rectum cancer** | | | | | | |
| Male | 6782.36(4093.52–10886.63) | 14353.38(9628.93–21562.66) | 111.63 | 123.75(75.26–195.68) | 136.11(92.55–203.91) | 9.99 |
| Female | 7029.17(4160.44–12448.75) | 15006.13(11377.86–19533.96) | 113.48 | 135.05(81.38–234.32) | 127.67(97.31–165.96) | -5.46 |
| Both | 13811.53(9319.29–21594.54) | 29359.51(23065.58–37900.19) | 112.57 | 129.41(88.98–199.74) | 131.97(104.79–170.15) | 1.98 |
| **Esophageal cancer** | | | | | | |
| Male | 9781.13(7616.07–12377.18) | 18205.52(13974.55–22559.74) | 86.13 | 171.61(135.55–215.03) | 166.35(128.53–204.37) | -3.07 |
| Female | 7944.11(6229.52–10392.1) | 9566.71(7282.53–12744.13) | 20.43 | 147.38(114.91–193.6) | 78.9(60.48–105.27) | -46.46 |
| Both | 17725.24(14876.69–21312.85) | 27772.23(22568.33–33243.15) | 56.68 | 160.15(135.32–192.8) | 120.84(98.29–144.14) | -24.54 |
| **Lip and oral cavity cancer** | | | | | | |
| Male | 12958.73(8493.57–19710.36) | 18106.19(12446.39–24249.03) | 39.72 | 217.24(142.82–327.23) | 162.31(112.74–216.52) | -25.29 |
| Female | 5288.79(3885.43–6968.44) | 10206.2(7943.2–12872.51) | 92.98 | 97.05(71.51–127.32) | 83.11(65.33–104.81) | -14.37 |
| Both | 18247.52(13430.9–25260.41) | 28312.39(22176.93–35115.35) | 55.16 | 158.85(116.1–217.88) | 121.06(95.67–149.37) | -23.79 |
| **Stomach cancer** | | | | | | |
| Male | 18200.42(13688.47–23212.23) | 22302(16520.03–28408.76) | 22.54 | 325.39(250.52–413.59) | 207.64(155.57–262.01) | -36.19 |
| Female | 14374.89(10749.33–20490.87) | 19827.9(15300.75–25419.27) | 37.93 | 250.58(189.4–358.48) | 160.19(125.48–205.26) | -36.07 |
| Both | 32575.32(26420.94–39913.48) | 42129.89(34853.01–49963.98) | 29.33 | 289.65(239.63–350.91) | 183.41(152.47–217.35) | -36.68 |
| **Bladder cancer** | | | | | | |
| Male | 1699.78(1227.29–2912.21) | 3771.12(2703.98–6547.77) | 121.86 | 35.86(25.98–59.38) | 38.36(27.84–65.81) | 6.96 |
| Female | 884.96(588.57–1333.67) | 1619.86(1145.6–2120.24) | 83.04 | 18.66(12.45–27.94) | 14.48(10.25–18.97) | -22.37 |
| Both | 2584.74(1959.37–3785.01) | 5390.97(4157.5–8115.11) | 108.57 | 27.4(20.76–39.17) | 25.76(19.9–38.36) | -6 |
| **Brain and nervous system cancer** | | | | | | |
| Male | 6263.41(3159.06–11555.85) | 9344.74(4421.34–17918.89) | 49.2 | 69.39(32.19–138.28) | 71.56(33.94–136.78) | 3.14 |
| Female | 9941.46(2906.21–18698.98) | 8913.47(5919.18–13908.98) | -10.34 | 100.62(33.76–186.79) | 61.95(40.92–96.12) | -38.43 |
| Both | 16204.87(9031.45–25820.6) | 18258.21(11886.93–30586.76) | 12.67 | 84.8(54.38–132.57) | 66.68(43.8–110.48) | -21.37 |
| **Gallbladder and biliary tract cancer** | | | | | | |
| Male | 2168.46(1503.83–3278.42) | 4877.02(2978.67–6941.07) | 124.91 | 41.03(29.04–62.42) | 46.33(28.63–65.73) | 12.92 |
| Female | 4060.99(2771.53–8058.47) | 9729.14(6830.38–14961.62) | 139.58 | 81.11(56.13–163.33) | 82.48(58.02–127.49) | 1.69 |
| Both | 6229.45(4597.35–10689.01) | 14606.16(10796.77–19727.16) | 134.47 | 60.54(45.07–104.51) | 65.29(48.72–87.48) | 7.83 |
| **Hodgkin lymphoma** | | | | | | |
| Male | 5364.56(3346.34–8725.41) | 3032.55(1696.8–6033.83) | -43.47 | 62.42(40.53–103.52) | 22.81(12.72–45.33) | -63.45 |

*(Continued)*

**Table 5.** (Continued)

| Morphology | All-Age DALYs (95% UI) | | | Age-standardized DALY Rate (95% UI), per 100 000 | | |
|---|---|---|---|---|---|---|
| | 1990 | 2017 | Change, % | 1990 | 2017 | Change, % |
| Female | 2366.09(1542.06–3382.96) | 1972.47(1112.79–3729.52) | -16.64 | 26.96(18.16–38.11) | 12.51(7.25–23.21) | -53.61 |
| Both | 7730.65(5123.25–11554.27) | 5005.03(2981.11–9214.02) | -35.26 | 44.88(31.49–67.27) | 17.49(10.5–32.11) | -61.02 |
| **Kidney cancer** | | | | | | |
| Male | 890.69(529.92–1551.53) | 2382.75(1499.7–3979.88) | 167.52 | 12.81(7.64–20.35) | 21.16(13.31–35.37) | 65.15 |
| Female | 764.09(445.22–1388.01) | 1381.76(911.47–2066.78) | 80.84 | 10.52(6.12–17.74) | 10.87(7.21–16.23) | 3.33 |
| Both | 1654.77(1054.5–2596.32) | 3764.51(2545.23–5691.44) | 127.49 | 11.69(7.32–16.72) | 15.81(10.74–23.79) | 35.22 |
| **Larynx cancer** | | | | | | |
| Male | 9307.9(7001.15–12217.29) | 12569.52(9415.08–16618.73) | 35.04 | 162.87(124.57–212.47) | 113.77(85.66–149.85) | -30.15 |
| Female | 4453.42(3138.75–5969.74) | 5338.77(3716.81–6974.43) | 19.88 | 78.55(53.96–104.58) | 42.88(29.58–56.12) | -45.41 |
| Both | 13761.32(11006.05–16856.67) | 17908.29(14393.78–21738.22) | 30.13 | 122.15(97.6–149.98) | 76.95(61.77–92.92) | -37 |
| **Leukemia** | | | | | | |
| Male | 13699.49(8828.52–19380.07) | 16139.7(10681.9–22110.92) | 17.81 | 143.59(102.45–186.9) | 121.06(79.82–163.54) | -15.69 |
| Female | 14346.11(8168.65–29003.8) | 16430.36(12013.03–22188.88) | 14.53 | 150.51(94.83–272.37) | 111.86(81.9–149) | -25.68 |
| Both | 28045.6(17665.61–46777.53) | 32570.06(24272.61–41539.81) | 16.13 | 147.07(106.65–213.95) | 116.62(87.74–148.12) | -20.71 |
| **Liver cancer** | | | | | | |
| Male | 4023.17(2354.9–5921.8) | 12408(7194.62–21419.52) | 208.41 | 68.71(39.88–99.5) | 112.56(64.98–192.54) | 63.83 |
| Female | 2381.26(1241–3409.39) | 5679.39(4003.37–7974.88) | 138.5 | 42.42(22.52–60.99) | 46.75(32.84–65.33) | 10.19 |
| Both | 6404.43(4594.46–8544.3) | 18087.38(12329.42–28502.49) | 182.42 | 55.98(40.39–74.52) | 78.25(53.82–122.81) | 39.8 |
| **Malignant skin melanoma** | | | | | | |
| Male | 394.73(249.67–724.18) | 637.22(403.59–1080.72) | 61.43 | 6.12(3.94–11.05) | 5.57(3.54–9.35) | -9 |
| Female | 278.1(137.31–708.35) | 606.26(356.73–1304.31) | 118.01 | 4.59(2.34–11.48) | 4.63(2.74–9.87) | 0.98 |
| Both | 672.82(436.95–1274.71) | 1243.48(821.69–2175.29) | 84.82 | 5.36(3.55–10.17) | 5.09(3.38–8.9) | -5.02 |
| **Mesothelioma** | | | | | | |
| Male | 250.9(130.09–506.1) | 579.5(369.57–836.58) | 130.96 | 4.33(2.26–8.64) | 5.31(3.42–7.59) | 22.55 |
| Female | 272.8(104.66–550.88) | 364.64(228.28–564.09) | 33.66 | 4.45(1.78–8.76) | 2.8(1.78–4.31) | -37.1 |
| Both | 523.71(274.35–1017.2) | 944.14(651.38–1365.7) | 80.28 | 4.41(2.34–8.45) | 4.01(2.79–5.74) | -9.01 |
| **Multiple myeloma** | | | | | | |
| Male | 958.81(560.32–1623.77) | 2357.43(1425.98–4003.06) | 145.87 | 16.74(9.91–28.87) | 21.35(13.01–36.02) | 27.53 |
| Female | 992.67(661.21–1542.75) | 2781.13(1830.01–4417.11) | 180.17 | 19.03(12.6–30.31) | 23.21(15.3–36.66) | 21.97 |
| Both | 1951.48(1343.81–3032.25) | 5138.56(3593.1–7850.55) | 163.32 | 17.87(12.52–27.71) | 22.35(15.74–33.83) | 25.08 |
| **Nasopharynx cancer** | | | | | | |
| Male | 2256.36(1574.74–3032.74) | 3376.97(2471.8–4382.12) | 49.66 | 36.12(25.82–48.08) | 29.56(21.6–38.33) | -18.17 |
| Female | 1876.59(1387.34–2418.6) | 2048.27(1537.15–2588.67) | 9.15 | 29.37(21.8–38.21) | 15.58(11.78–19.73) | -46.95 |
| Both | 4132.95(3303.98–5015.21) | 5425.24(4369.06–6514.43) | 31.27 | 32.87(26.6–39.56) | 22.28(18.01–26.57) | -32.2 |
| **Non-Hodgkin lymphoma** | | | | | | |
| Male | 4582.68(2943.31–7372.61) | 7723.88(5336.26–11423.65) | 68.55 | 60.87(38.72–91.33) | 64.28(43.86–95.22) | 5.61 |
| Female | 2809.8(1883.51–4068.61) | 5758.65(3446.67–8207.41) | 104.95 | 38.54(26.53–53.42) | 43.67(26.43–62.69) | 13.3 |
| Both | 7392.48(5423.58–10133.57) | 13482.53(10024.23–17228.9) | 82.38 | 49.8(38.86–64.35) | 53.6(40.27–68.56) | 7.64 |
| **Non-melanoma skin cancer** | | | | | | |
| Male | 1296.76(801.1–1746.42) | 2512.27(1626.13–3224.6) | 93.73 | 25.54(15.2–34.26) | 24.93(16.02–31.72) | -2.41 |
| Female | 283.18(170.89–489.22) | 551.88(424.05–724.35) | 94.89 | 5.63(3.62–9.34) | 4.83(3.72–6.27) | -14.17 |
| Both | 1579.94(1055.83–2096.11) | 3064.16(2125.64–3821.52) | 93.94 | 15.77(10.22–21.04) | 14.31(9.81–17.8) | -9.23 |
| **Other pharynx cancer** | | | | | | |
| Male | 6415.08(3672.12–10137.04) | 13683.54(9201.5–19317.98) | 113.3 | 108.72(62.45–171.18) | 122.49(82.42–172.31) | 12.67 |
| Female | 3988.83(2964.75–5463.99) | 7604.4(5708.45–10527.05) | 90.64 | 69.44(51.49–94.79) | 60.44(45.46–83.5) | -12.95 |
| Both | 10403.9(7710.28–14520.96) | 21287.94(16071.92–27968.49) | 104.61 | 89.84(65.99–126.09) | 90.32(68.26–118.22) | 0.53 |

(*Continued*)

**Table 5.** (Continued)

| Morphology | All-Age DALYs (95% UI) | | | Age-standardized DALY Rate (95% UI), per 100 000 | | |
|---|---|---|---|---|---|---|
| | 1990 | 2017 | Change, % | 1990 | 2017 | Change, % |
| **Ovarian cancer** | | | | | | |
| Female | 3078.23(2072.9–5495.34) | 9405.66(6937.78–12473.28) | 205.55 | 53.66(36.49–93.9) | 74.41(54.99–98.26) | 38.68 |
| Both | 3078.23(2072.9–5495.34) | 9405.66(6937.78–12473.28) | 205.55 | 26.19(17.81–45.78) | 39.1(28.95–51.52) | 49.32 |
| **Pancreatic cancer** | | | | | | |
| Male | 2125.43(1364.4–3433.56) | 6929.6(4343.53–11245.98) | 226.03 | 38.75(24.84–61.94) | 65.12(41.8–104.65) | 68.02 |
| Female | 1467.51(997.25–1995.03) | 5898.89(3933.24–7910.21) | 301.97 | 29.76(19.47–40.58) | 50.54(33.57–67.81) | 69.81 |
| Both | 3592.93(2499.74–5052.19) | 12828.49(8887.95–18122.03) | 257.05 | 34.37(23.46–48.05) | 57.54(39.95–80.98) | 67.4 |
| **Prostate cancer** | | | | | | |
| Male | 4054.97(2943.3–5515.14) | 11072.92(7848.86–14710.17) | 173.07 | 100.17(71.05–136.33) | 121.18(85.96–159.38) | 20.97 |
| Both | 4054.97(2943.3–5515.14) | 11072.92(7848.86–14710.17) | 173.07 | 50.63(35.94–68.86) | 56.75(40.28–74.87) | 12.08 |
| **Testicular cancer** | | | | | | |
| Male | 1577.81(951.64–2218.72) | 907.02(538.54–1280.61) | -42.51 | 19.34(11.98–27.02) | 6.7(3.84–9.42) | -65.34 |
| Both | 1577.81(951.64–2218.72) | 907.02(538.54–1280.61) | -42.51 | 9.45(5.88–13.2) | 3.06(1.77–4.28) | -67.62 |
| **Thyroid cancer** | | | | | | |
| Male | 696.64(483.45–967.48) | 1367.83(990.21–1807.24) | 96.35 | 10.93(7.71–14.84) | 11.98(8.75–15.78) | 9.54 |
| Female | 1547.42(1020.32–2607.5) | 2545.31(1775.4–4189.15) | 64.49 | 23.09(15.4–40.11) | 18.89(13.38–30.63) | -18.2 |
| Both | 2244.06(1624.76–3276.11) | 3913.14(3028.83–5290.95) | 74.38 | 17.02(12.72–24.78) | 15.69(12.35–20.97) | -7.77 |
| **Uterine cancer** | | | | | | |
| Female | 2426.67(1372.55–3449.38) | 3637.86(2593.66–4938.2) | 49.91 | 46.41(27.01–64.63) | 30.17(21.43–40.52) | -34.99 |
| Both | 2426.67(1372.55–3449.38) | 3637.86(2593.66–4938.2) | 49.91 | 22.58(13.15–31.41) | 15.79(11.24–21.2) | -30.09 |

smoking tobacco; and tobacco is sold widely in cheap brands [21, 22]. World Health Organization's STEPwise approach to non-communicable disease risk factor surveillance (STEPS) survey in 2019 showed that 28.0% of men and 7.5% of women smoked tobacco products; 33.3% of

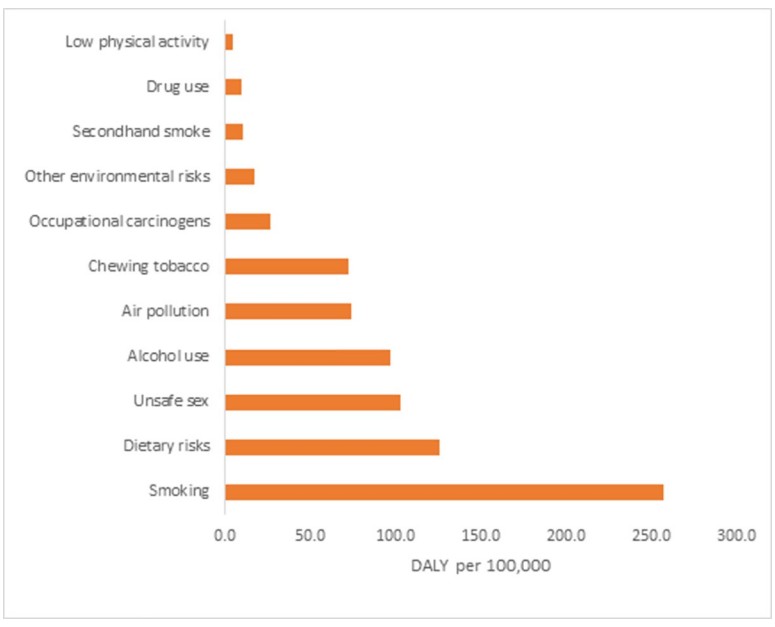

**Fig 2. Risk factors of cancer contributing to DALYs in Nepal, 2017.**

men and 4.9% of women used smokeless tobacco; and that the prevalence of smoking among adult Nepalese females was one of the highest in the WHO South-East Asia Region [23]. In a survey of the national burden of disease in 2017, almost 13% of deaths are attributable to smoking [24]. Nepal ratified the WHO Framework Convention on Tobacco Control (WHO FCTC) in 2006 and formulated National Anti-Tobacco Communication Campaign Strategy. The government has formulated a tobacco control law, utilized mass media for health education, an excise tax on tobacco, banned smoking in public places, and made efforts to reduce the supply of tobacco products [25]. The high burden of lung cancer in women despite less burden of smoking could be attributed to the inhalation of indoor smoke, and second-hand smoke among rural women [10].

## Breast cancer and cervical cancer

Among females, cancer is the second leading cause of death worldwide with breast, colorectal, and lung cancers leading the charts [26]. But even globally, between 2007 and 2017, cervical cancer incidence and mortality rates have increased by 19% each [27]. This contrasts with our data where the second most common cancer in the female is cervical cancer. Female-related cancers, namely, breast cancer followed by cervical cancer have an increasing trend of age-standardized incidence in Nepal. In our study, cervical cancer has a decreasing trend of incidence among females. The national agenda to curb diseases such as cervical cancer and breast cancer might have potentially played a role.

Early detection of breast cancer is available through awareness and screening. However, coverage of mammography is still limited, and most women are diagnosed in a late stage leading to suboptimal survival. Despite ongoing public health efforts, there is low knowledge of breast cancer among Nepalese women [28]. Factors that contribute to an increasing trend in breast cancer incidence among Asian women are not fully understood but thought to reflect lifestyle changes associated with westernization, including late childbearing, having fewer children, and consumption of calorie-dense food, physical inactivity, and obesity [29]. The increasing trend in our study may reflect a collaboration of changed environmental factors, including the delay of childbearing, increases in the levels of obesity, and early cancer screening.

The higher burden of cervical cancer among Nepalese women is due to Human papillomavirus (HPV) infection. In developing countries like Nepal, women belonging to lower socioeconomic status higher levels of illiteracy are uncomfortable sharing the symptoms like abnormal vaginal bleeding such as post-coital bleeding, intermenstrual bleeding, or post-menopausal bleeding immediately after onset leading to late diagnosis [30]. However, the increased incidence and decreased trend in mortality can be credited to the cervical cancer screening programs leading to its early detection and treatment. Visual Inspection with acetic acid as recommended by WHO in low-income countries for the early detection of cervical cancer has been advocated in the public health system of Nepal making the procedure freely available from the health post to the tertiary care center [31]. Nepal has an intermediate burden of HPV infection and theoretically, approximately 80% of cervical cancer is preventable by HPV vaccines [32]. There have been a few pilot programs for the demonstration of the HPV vaccine in some districts of Nepal [33, 34]. But still, the HPV vaccine is not available in the National Immunization Program.

## Stomach cancer

Stomach cancer ranks fourth in the national cancer burden. Stomach cancer has a decreasing trend of incidence in both males and females. Dietary patterns also ranked the third risk factor

associated with DALY. The decreasing trend of incidence and prevalence of this cancer can be explained by the increased availability of upper gastro-intestinal endoscopic screening leading to early detection and management and hence higher survival rates of stomach cancer. It is known that apart from smoking, *Helicobacter pylori* infection is a known risk factor for stomach cancer which was found to be around 16% among the study population that increased among the lower socio-economic population as found by Ansari et al. in their hospital-based study [35]. There is a decreasing prevalence of *H. pylori* infection due to highly effective anti-microbial therapies; cheaper and sensitive laboratory tests like serum antigen detection and *H. pylori* urea breath testing; improved living conditions, and healthy and hygienic food practices lead to the declining rates.

## Oral cavity cancer

In our study, there is a decreasing trend in the incidence of oral cavity cancer in both sexes. It is the second most common cancer among men in South-Central Asia [36]. This region has high incidence rates for oral cancer because, in addition to tobacco smoking, tobacco chewing as well as chewing betel quid and areca nut also poses a major risk in acquiring oral cavity cancer. In a study, tobacco consumption and alcohol drinking were responsible for almost 85.3% of head and neck cancers with a population attributable fraction (PAF) of 24.3% for smoking, 39.9% for tobacco chewing, and 23.0% for alcohol drinking [37]. The interplay of the trends of the two risk factors to which the highest proportion of cancer DALYs in Nepal could be attributed to tobacco and alcohol use; and their consumption in Nepal has increased during this period [9].

## Colorectal cancer (CRC)

Globally, while there is a declining trend of CRC incidence worldwide, there is an increasing trend of incidence and mortality for CRC in Asian countries [38]. In this study, there is also an increased incidence of CRC among both sexes in Nepal. Similar to our findings, the other studies conducted in Tribhuvan University Teaching Hospital (Kathmandu, Nepal) from 1990 to 2008 and in B.P. Koirala Memorial Cancer Hospital (Chitwan, Nepal) from 2014 to 2018 also reported an increasing proportion of younger age groups in CRC incidence with slight male preponderance [39, 40]. Globally, the declining trend of CRC has largely been associated with an increase in screening rates in 50 years or older age groups but the incidence rates are increasing among adults under 50 years, for whom screening is not recommended [41].

Unfavorable increased colorectal cancer rates are thought to reflect changes in dietary patterns, obesity, and smoking rates, often seen in economically transitioning countries. Apart from predominant modifiable risk factors like tobacco and alcohol consumption in Nepal, a poor diet like low consumption of fruits and vegetables which are rich in fiber; and high consumption of red or processed meat ranks significantly high in contributing to DALY [42]. With public health strategies addressing to make modest changes in the consumption of alcohol and red-processed meat, weight loss and increased levels of physical activity may translate into significant reductions in the incidence of colorectal cancer [43].

## Other cancers

In our study, among the male-related cancers, there is a rapid rise in prostate cancer incidence and a decrease in the incidence of testicular cancer. Worldwide in 2017, prostate cancer had the highest incidence among men in 114 countries and was the fifth leading cause of death from cancer among men in 56 countries [27]. In contrast to the declining incidence of prostate cancer in Western countries, the rates have increased in some Asian countries including India

[44]. Differences in average life expectancy, Western foods high in calories and fat, and medical checkups for prostate cancer in Asian countries, may explain the difference in incidence in different countries. The increasing incidence rates as detected by efficient use of the combination of the digital rectal examination, serum prostate-specific antigen, and transrectal ultrasonographic evaluation with systematic biopsy, combined with an aging and growing population has led to an increase in prostate cancer cases since 2007 [27]. Testicular cancer is the most common cancer among young men between ages 20–34 in Asian countries with a high Human Development Index (HDI) with a positive correlation between HDI and the standardized incidence rate of testicular cancer and negative correlation with standardized mortality rate [45]. Nepal belongs to the medium human development category, positioning it at 147 out of 189 countries and territories where testicular cancers don't comprise a significant burden and have a decreasing trend.

There is very few oncology dedicated tertiary care hospitals in Nepal with health care experts. For a country with a 28.09 million population, the reach of comprehensive and affordable oncology-specific tertiary care is still sparse. It is estimated that the average direct cost of cancer care in Nepal is NRs 387,000 (1 USD = NRs 119) and the average medical cost is NRs 313,000 which is higher than the average annual income of a Nepalese citizen (NRs 78,946.00) [6]. The Government of Nepal provides a fund of only US$ 1000 (NRs 100,000) to support patients with cancer [46], which is not enough and many non-government organizations are working on improving cancer awareness and prevention in Nepal as well [47].

## Recommendations

Nepal's trend of cancer incidence and mortality have increased in 27 years with fluctuations in some cancers trend. Strong implementation of the tobacco control law is needed along with an increase in tobacco excise taxes and tobacco cessation programs. Restriction or increase of the tax for processed foods seems to be an important step in reducing diet-related cancers. National Cancer Registry is a landmark initiative by the Government of Nepal established in 2003 to generate data on the enormity and trends of cancer through the population and hospital-based registries [48]. However, these data limit information on the patterns of cancer burden and epidemiology in Nepal. A robust cancer registry, production of oncology-related human resources, and development of diagnostic and treatment facilities, along with the incorporation of HPV vaccine in the national immunization program can be the other modalities for the prevention of cancers. Creating awareness on cancer among the public along with the importance of physical activity and diet and screening for early diagnosis and treatment is fundamental in reducing the future trend. The fund provided by the government should also be increased to address the financial burden to cancer patients and hence help increase the quality of life.

## Strength and limitations

This is the first unique report that provides the most comprehensive estimates on trends and distribution of the burden of cancer at a national level in Nepal from 1990 to 2017. However, a few vital limitations should also be considered. First, there is a lack of primary data sources due to limited resources in Nepal. Second, the data used for this study were derived from GBD 2017, so all the general limitations attributed to that study's methodology are also applicable here [16, 49, 50]. Third, the assessment of the burden of cancer was restricted to standard epidemiological parameters, meaning monetary and social burdens were not considered. These limitations may signify this study undervalued the actual cancer burden in Nepal.

## Conclusions

This study highlighted the burden of different types of cancer in Nepal over 27 years. Cancer is a major public health problem and accounted for 10% of total deaths in Nepal. The incidence and mortality due to cancer are in an increasing trend with a high impact on DALYs. Breast followed by lung, cervical, stomach and oral cavity cancers were the topmost cancers. Tobacco use, unhealthy food, and unsafe sexual behaviors are the predominant risk factors for cancer. This calls for urgent measures to raise awareness by health education intervention and implement effective cancer screening programs all over the country.

## Author Contributions

**Conceptualization:** Gambhir Shrestha, Rahul Kumar Thakur, Rajshree Singh, Rashmi Mulmi, Pranil Man Singh Pradhan.

**Formal analysis:** Gambhir Shrestha, Rahul Kumar Thakur, Rajshree Singh, Rashmi Mulmi, Abha Shrestha.

**Methodology:** Gambhir Shrestha, Rahul Kumar Thakur, Rajshree Singh.

**Project administration:** Gambhir Shrestha.

**Software:** Gambhir Shrestha, Rahul Kumar Thakur, Rajshree Singh.

**Supervision:** Gambhir Shrestha, Pranil Man Singh Pradhan.

**Validation:** Gambhir Shrestha, Rajshree Singh, Abha Shrestha, Pranil Man Singh Pradhan.

**Visualization:** Rajshree Singh, Rashmi Mulmi, Abha Shrestha, Pranil Man Singh Pradhan.

**Writing – original draft:** Gambhir Shrestha, Rahul Kumar Thakur, Rajshree Singh, Rashmi Mulmi, Abha Shrestha, Pranil Man Singh Pradhan.

**Writing – review & editing:** Gambhir Shrestha, Rahul Kumar Thakur, Rajshree Singh, Rashmi Mulmi, Abha Shrestha, Pranil Man Singh Pradhan.

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
