## [Decision Letter · Decision Letter 0]

24 Jun 2021

PONE-D-21-04298

Cancer burden in Nepal, 1990-2017: An analysis of the Global Burden of Disease Study

PLOS ONE

Dear Dr. Shrestha,

Thank you for submitting your manuscript to PLOS ONE. After careful consideration, we feel that it has merit but does not fully meet PLOS ONE’s publication criteria as it currently stands. Therefore, we invite you to submit a revised version of the manuscript that addresses the points raised during the review process.

We look forward to receiving your revised manuscript.

Kind regards,

Rashidul Alam Mahumud, MPH, MSc, PhD

Academic Editor

PLOS ONE

Journal Requirements:

Reviewers' comments:

Reviewer's Responses to Questions

**Comments to the Author**

1. Is the manuscript technically sound, and do the data support the conclusions?

Reviewer #1: Yes

2. Has the statistical analysis been performed appropriately and rigorously? 

Reviewer #1: Yes

3. Have the authors made all data underlying the findings in their manuscript fully available?

Reviewer #1: Yes

4. Is the manuscript presented in an intelligible fashion and written in standard English?

Reviewer #1: Yes

5. Review Comments to the Author

Reviewer #1: Generally, it is well organised research paper, finding are interesting but there are some issues need to address;

1. Abstract should match with the full text; you said WHO and IHME database in the main text but not clear in the abstract.

2. Line 43 try to make a complex sentence.

3. Line 45 what is GLOBOCAN??

4. Line 50-53 need references

5. Line 67-69 no need to inform it is free, rewrite this sentence

6. Line 82 Where you used the WHO data? Please explain details regarding it.

7. Line 91 not needed

8. Line 210 Start discussion from the beginning of your study findings

9. Line 211-222 Did your findings support to discuss this?

10. Line 254-258 need references

11. Line 299-300 Make it clear

12. Line 305 Need references

13. Line 329-330 Which study and when?

14. Line 385 Conclusion; need to add more based on the findings

6. PLOS authors have the option to publish the peer review history of their article (what does this mean?). If published, this will include your full peer review and any attached files.

Reviewer #1: **Yes: **Padam Dahal

---

## [Author Response · Author response to Decision Letter 0]

29 Jun 2021

Author’s responses to editor and reviewers’ comments/suggestions

Editor

1. Editor’s comment: Please ensure that your manuscript meets PLOS ONE's style requirements, including those for file naming.

Authors’ reply: We have revised the manuscript as per the journal’s style requirements.

2. Editor’s comment: Please review your reference list to ensure that it is complete and correct.

Authors’ reply: We have reviewed the entire reference list and ensured that it is complete and correct.

We do not have any retracted papers cited in this manuscript.

Reviewer 1

1. Reviewer’s comment: Abstract should match with the full text; you said WHO and IHME database in the main text but not clear in the abstract.

Authors’ reply: Thank you for picking up this point. We have removed WHO as we have used only IHME’s GBD data in our study. We have also revised the abstract to match the full text (revised manuscript page number 2). 

2. Reviewer’s comment: Line 43 try to make a complex sentence.

Authors’ reply: We have revised the lines 43 and 44 in a single sentence (revised manuscript page number 4, line 48,49).

3. Reviewer’s comment: Line 45 what is GLOBOCAN??

Authors’ reply: We have used the full form Global Cancer Observatory, which annually provides data on cancer for all countries (revised manuscript page number 4 line 49)

4. Reviewer’s comment: Line 50-53 need references

Authors’ reply: References added to the sentence (revised manuscript page number 4, line 55-58).

5. Reviewer’s comment: Line 67-69 no need to inform it is free, rewrite this sentence

Authors’ reply: We have removed the “freely available” phrase and rewrote the sentence (revised manuscript page number 6, line 82-84).

6. Reviewer’s comment: Line 82 Where you used the WHO data? Please explain details regarding it.

Authors’ reply: Thank you for this query. We have not used the WHO data. It is now corrected in the manuscript (Revised manuscript page number 6, line 82-88).

7. Reviewer’s comment: Line 91 not needed

Authors’ reply: We have deleted the ethical approval part from the manuscript as suggested.

8. Reviewer’s comment: Line 210 Start discussion from the beginning of your study findings

Authors’ reply: We have now revised the discussion by mentioning the major findings of the study in the beginning (Revised manuscript page number 34, line 242-246).

9. Reviewer’s comment: Line 211-222 Did your findings support to discuss this? 

Authors’ reply: We have shifted this paragraph to the end of the discussion to make sense as well as to inform the readers about the availability of treatment centers in Nepal and the government support to the cancer patients (Revised manuscript page number 40-41, line 390-398).

10. Reviewer’s comment: Line 254-258 need references

Authors’ reply: We have added the references for this sentence (Revised Manuscript page number 36, line 270-274).

11. Reviewer’s comment: Line 299-300 Make it clear

Authors’ reply: We have rewritten the sentence to make it clear as " But still the HPV vaccine is not available in the National Immunization Program. (Revised manuscript page number 37, line 322-323).

12. Reviewer’s comment: Line 305 Need references

Authors’ reply: This is the findings from this study and hence this sentence will not require the reference (revised line 325)

13. Reviewer’s comment: Line 329-330 Which study and when?

Authors’ reply: We have mentioned the names of the institution with address and the period when the studies were conducted in the revised manuscript (Revised manuscript page number 39, line 354-357).

14. Reviewer’s comment: Line 385 Conclusion; need to add more based on the findings

Authors’ reply: We have revised the conclusion based on our findings (revised page number 43).

---

## [Editor Report · Decision Letter 1]

19 Jul 2021

Cancer burden in Nepal, 1990-2017: an analysis of the Global Burden of Disease Study

PONE-D-21-04298R1

Dear Dr. Shrestha,

We’re pleased to inform you that your manuscript has been judged scientifically suitable for publication and will be formally accepted for publication once it meets all outstanding technical requirements.

Kind regards,

Rashidul Alam Mahumud, MPH, MSc, PhD

Academic Editor

PLOS ONE
---

## [Editor Report · Acceptance letter]

26 Jul 2021

PONE-D-21-04298R1 

Cancer burden in Nepal, 1990-2017: an analysis of the Global Burden of Disease Study 

Dear Dr. Shrestha:

I'm pleased to inform you that your manuscript has been deemed suitable for publication in PLOS ONE. Congratulations! Your manuscript is now with our production department. 

Kind regards, 

on behalf of

Dr. Rashidul Alam Mahumud 

Academic Editor

PLOS ONE